# Multimodal Generative Learning Utilizing Jensen-Shannon-Divergence

**Thomas M. Sutter,   Imant Daunhawer,   Julia E. Vogt**
Department of Computer Science
ETH Zurich
{thomas.sutter,imant.daunhawer,julia.vogt}@inf.ethz.ch

## Abstract

Learning from different data types is a long-standing goal in machine learning research, as multiple information sources co-occur when describing natural phenomena. However, existing generative models that approximate a multimodal ELBO rely on difficult or inefficient training schemes to learn a joint distribution and the dependencies between modalities. In this work, we propose a novel, efficient objective function that utilizes the Jensen-Shannon divergence for multiple distributions. It simultaneously approximates the unimodal and joint multimodal posteriors directly via a dynamic prior. In addition, we theoretically prove that the new multimodal JS-divergence (mmJSD) objective optimizes an ELBO. In extensive experiments, we demonstrate the advantage of the proposed mmJSD model compared to previous work in unsupervised, generative learning tasks.

## 1   Introduction

Replicating the human ability to process and relate information coming from different sources and learn from these is a long-standing goal in machine learning [2]. Multiple information sources offer the potential of learning better and more generalizable representations, but pose challenges at the same time: models have to be aware of complex intra- and inter-modal relationships, and be robust to missing modalities [18, 31]. However, the excessive labelling of multiple data types is expensive and hinders possible applications of fully-supervised approaches [6, 11]. Simultaneous observations of multiple modalities moreover provide self-supervision in the form of shared information which connects the different modalities. Self-supervised, generative models are a promising approach to capture this joint distribution and flexibly support missing modalities with no additional labelling cost attached. Based on the shortcomings of previous work (see Section 2.1), we formulate the following wish-list for multimodal, generative models:

**Scalability.** The model should be able to efficiently handle any number of modalities. Translation approaches [10, 32] have had great success in combining two modalities and translating from one to the other. However, the training of these models is computationally expensive for more than two modalities due to the exponentially growing number of possible paths between subsets of modalities.

**Missing data.** A multimodal method should be robust to missing data and handle any combination of available and missing data types. For discriminative tasks, the loss in performance should be minimized. For generation, the estimation of missing data types should be conditioned on and coherent with available data while providing diversity over modality-specific attributes in the generated samples.

**Information gain.** Multimodal models should benefit from multiple modalities for discriminative as well as for generative tasks.

In this work, we introduce a novel probabilistic, generative and self-supervised multi-modal model. The proposed model is able to integrate information from different modalities, reduce uncertainty and ambiguity in redundant sources, as well as handle missing modalities while making no assumptions about the nature of the data, especially about the inter-modality relations.

We base our approach directly in the Variational Bayesian Inference framework and propose the new multimodal Jensen-Shannon divergence (mmJSD) objective. We introduce the idea of a dynamic prior for multimodal data, which enables the use of the Jensen-Shannon divergence for $M$ distributions [1, 15] and interlinks the unimodal probabilistic representations of the $M$ observation types. Additionally, we are - to the best of our knowledge - the first to empirically show the advantage of modality-specific subspaces for multiple data types in a self-supervised and scalable setting. For the experiments, we concentrate on Variational Autoencoders [12]. In this setting, our multimodal extension to variational inference implements a scalable method, capable of handling missing observations, generating coherent samples and learning meaningful representations. We empirically show this on two different datasets. In the context of scalable generative models, we are the first to perform experiments on datasets with more than 2 modalities showing the ability of the proposed method to perform well in a setting with multiple modalities.

## 2 Theoretical Background & Related Work

We consider some dataset of $N$ i.i.d. sets $\{\boldsymbol{X}^{(i)}\}_{i=1}^{N}$ with every $\boldsymbol{X}^{(i)}$ being a set of $M$ modalities $\boldsymbol{X}^{(i)} = \{\boldsymbol{x}_j^{(i)}\}_{j=1}^{M}$. We assume that the data is generated by some random process involving a joint hidden random variable $\boldsymbol{z}$ where inter-modality dependencies are unknown. In general, the same assumptions are valid as in the unimodal setting [12]. The marginal log-likelihood can be decomposed into a sum over marginal log-likelihoods of individual sets $\log p_\theta(\{\boldsymbol{X}^{(i)}\}_{i=1}^{N}) = \sum_{i=1}^{N} \log p_\theta(\boldsymbol{X}^{(i)})$, which can be written as:

$$\log p_\theta(\boldsymbol{X}^{(i)}) = KL(q_\phi(\boldsymbol{z}|\boldsymbol{X}^{(i)})||p_\theta(\boldsymbol{z}|\boldsymbol{X}^{(i)})) + \mathcal{L}(\theta, \phi; \boldsymbol{X}^{(i)}), \tag{1}$$

$$\text{with } \mathcal{L}(\theta, \phi; \boldsymbol{X}^{(i)}) := E_{q_\phi(\boldsymbol{z}|\boldsymbol{X})}[\log p_\theta(\boldsymbol{X}^{(i)}|\boldsymbol{z})] - KL(q_\phi(\boldsymbol{z}|\boldsymbol{X}^{(i)})||p_\theta(\boldsymbol{z})). \tag{2}$$

$\mathcal{L}(\theta, \phi; \boldsymbol{X}^{(i)})$ is called evidence lower bound (ELBO) on the marginal log-likelihood of set $i$. The ELBO forms a computationally tractable objective to approximate the joint data distribution $\log p_\theta(\boldsymbol{X}^{(i)})$ which can be efficiently optimized, because it follows from the non-negativity of the KL-divergence: $\log p_\theta(\boldsymbol{X}^{(i)}) \geq \mathcal{L}(\theta, \phi; \boldsymbol{X}^{(i)})$. Particular to the multimodal case is what happens to the ELBO formulation if one or more data types are missing: we are only able to approximate the true posterior $p_\theta(\boldsymbol{z}|\boldsymbol{X}^{(i)})$ by the variational function $q_{\phi_K}(\boldsymbol{z}|\boldsymbol{X}_K^{(i)})$. $\boldsymbol{X}_K^{(i)}$ denotes a subset of $\boldsymbol{X}^{(i)}$ with $K$ available modalities where $K \leq M$. However, we would still like to be able to approximate the true multimodal posterior distribution $p_\theta(\boldsymbol{z}|\boldsymbol{X}^{(i)})$ of all data types. For simplicity, we always use $\boldsymbol{X}_K^{(i)}$ to symbolize missing data for set $i$, although there is no information about which or how many modalities are missing. Additionally, different modalities might be missing for different sets $i$. In this case, the ELBO formulation changes accordingly:

$$\mathcal{L}_K(\theta, \phi_K; \boldsymbol{X}^{(i)}) := E_{q_{\phi_K}(\boldsymbol{z}|\boldsymbol{X}_K^{(i)})}[\log(p_\theta(\boldsymbol{X}^{(i)}|\boldsymbol{z})] - KL(q_{\phi_K}(\boldsymbol{z}|\boldsymbol{X}_K^{(i)})||p_\theta(\boldsymbol{z})) \tag{3}$$

$\mathcal{L}_K(\theta, \phi_K; \boldsymbol{X}^{(i)})$ defines the ELBO if only $\boldsymbol{X}_K^{(i)}$ is available, but we are interested in the true posterior distribution $p_\theta(\boldsymbol{z}|\boldsymbol{X}^{(i)})$. To improve readability, we will omit the superscript $(i)$ in the remaining part of this work.

### 2.1 Related Work

In this work, we focus on methods with the aim of modelling a joint latent distribution, instead of translating between modalities [10, 24] due to the scalability constraint described in Section 1.

**Joint and Conditional Generation.** [23] implemented a multimodal VAE and introduced the idea that the distribution of the unimodal approximation should be close to the multimodal approximation function. [27] introduced the triple ELBO as an additional improvement. Both define labels as second modality and are not scalable in the number of modalities.

**Modality-specific Latent Subspaces.** [9, 26] both proposed models with modality-specific latent distributions and an additional shared distribution. The former relies on supervision by labels to extract modality-independent factors, while the latter is non-scalable. [5] are also able to show the advantage of modality-specific sub-spaces.

**Scalability.** More recently, [13, 29] proposed scalable multimodal generative models for which they achieve scalability by using a Product-of-Experts (PoE) [8] as a joint approximation distribution. The PoE allows them to handle missing modalities without requiring separate inference networks for every combination of missing and available data. A PoE is computationally attractive as - for Gaussian-distributed experts - it remains Gaussian distributed which allows the calculation of the KL-divergence in closed form. However, they report problems in optimizing the unimodal variational approximation distributions due to the multiplicative nature of the PoE. To overcome this limitation, [29] introduced a combination of ELBOs which results in the final objective not being an ELBO anymore [30]. [22] use a Mixture-of-Experts (MoE) as joint approximation function. The additive nature of the MoE facilitates the optimization of the individual experts, but is computationally less efficient as there exists no closed form solution to calculate the KL-divergence. [22] need to rely on importance sampling (IS) to achieve the desired quality of samples. IS based VAEs [4] tend to achieve tight ELBOs for the price of a reduced computational efficiency. Additionally, their model requires $M^2$ passes through the decoder networks which increases the computational cost further.

## 3 The multimodal JS-Divergence model

We propose a new multimodal objective (mmJSD) utilizing the Jensen-Shannon divergence. Compared to previous work, this formulation does not need any additional training objectives [29], supervision [26] or importance sampling [22], while being scalable [9].

**Definition 1.**

1. *Let $\boldsymbol{\pi}$ be the distribution weights: $\boldsymbol{\pi} = [\pi_1, \ldots, \pi_{M+1}]$ and $\sum_i \pi_i = 1$.*

2. *Let $JS_{\boldsymbol{\pi}}^{M+1}$ be the Jensen-Shannon divergence for $M+1$ distributions*

$$JS_{\boldsymbol{\pi}}^{M+1}(\{q_j(\boldsymbol{z})\}_{j=1}^{M+1}) = \sum_{j=1}^{M+1} \pi_j KL(q_j(\boldsymbol{z})|f_{\mathcal{M}}(\{q_\nu(\boldsymbol{z})\}_{\nu=1}^{M+1})). \tag{4}$$

   *where the function $f_{\mathcal{M}}$ defines a mixture distribution of its arguments [15].*

*We define a new objective $\widetilde{\mathcal{L}}(\theta, \phi; \boldsymbol{X})$ for learning multimodal, generative models which utilizes the Jensen-Shannon divergence:*

$$\widetilde{\mathcal{L}}(\theta, \phi; \boldsymbol{X}) := E_{q_\phi(\boldsymbol{z}|\boldsymbol{X})}[\log p_\theta(\boldsymbol{X}|\boldsymbol{z})] - JS_{\boldsymbol{\pi}}^{M+1}(\{q_{\phi_{\boldsymbol{z}_j}}(\boldsymbol{z}|\boldsymbol{x}_j)\}_{j=1}^M, p_\theta(\boldsymbol{z})) \tag{5}$$

The JS-divergence for $M+1$ distributions is the extension of the standard JS-divergence for two distributions to an arbitrary number of distributions. It is a weighted sum of KL-divergences between the $M+1$ individual probability distributions $q_j(\boldsymbol{z})$ and their mixture distribution $f_{\mathcal{M}}$. In the remaining part of this section, we derive the new objective directly from the standard ELBO formulation and prove that it is a lower bound to the marginal log-likelihood $\log p_\theta(\boldsymbol{X}^{(i)})$.

### 3.1 Joint Distribution

A MoE is an arithmetic mean function whose additive nature facilitates the optimization of the individual experts compared to a PoE (see Section 2.1). As there exists no closed form solution for the calculation of the respective KL-divergence, we need to rely on an upper bound to the true divergence using Jensen's inequality [7] for an efficient calculation (for details please see Appendix B). In a first step towards Equation (5), we approximate the multimodal ELBO defined in Equation (2) by a sum of KL-terms:

$$\mathcal{L}(\theta, \phi; \boldsymbol{X}) \geq E_{q_\phi(\boldsymbol{z}|\boldsymbol{X})}[\log p_\theta(\boldsymbol{X}|\boldsymbol{z})] - \sum_{j=1}^M \pi_j KL(q_{\phi_j}(\boldsymbol{z}|\boldsymbol{x}_j)||p_\theta(\boldsymbol{z})) \tag{6}$$

The sum of KL-divergences can be calculated in closed form if prior distribution $p_\theta(\boldsymbol{z})$ and unimodal posterior approximations $q_{\phi_j}(\boldsymbol{z}|\boldsymbol{x}_j)$ are both Gaussian distributed. In the Gaussian case, this lower bound to the ELBO $\mathcal{L}(\theta, \phi; \boldsymbol{X})$ allows the optimization of the ELBO objective in a computationally efficient way.

## 3.2 Dynamic Prior

In the regularization term in Equation (6), although efficiently optimizable, the unimodal approximations $q_{\phi_j}(\boldsymbol{z}|\boldsymbol{x}_j)$ are only individually compared to the prior, and no joint objective is involved. We propose to incoporate the unimodal posterior approximations into the prior through a function $f$.

**Definition 2** (Multimodal Dynamic Prior). *The dynamic prior is defined as a function $f$ of the unimodal approximation functions $\{q_{\phi_\nu}(\boldsymbol{z}|\boldsymbol{x}_\nu)\}_{\nu=1}^M$ and a pre-defined distribution $p_\theta(\boldsymbol{z})$:*

$$p_f(\boldsymbol{z}|\boldsymbol{X}) = f(\{q_{\phi_\nu}(\boldsymbol{z}|\boldsymbol{x}_\nu)\}_{\nu=1}^M, p_\theta(\boldsymbol{z})) \tag{7}$$

The dynamic prior is not a prior distribution in the conventional sense as it does not reflect prior knowledge of the data, but it incorporates the prior knowledge that all modalities share common factors. We therefore call it *prior* due to its role in the ELBO formulation and optimization. As a function of all the unimodal posterior approximations, the dynamic prior extracts the shared information and relates the unimodal approximations to it. With this formulation, the objective is optimized at the same time for a similarity between the function $f$ and the unimodal posterior approximations. For random sampling, the pre-defined prior $p_\theta(\boldsymbol{z})$ is used.

## 3.3 Jensen-Shannon Divergence

Utilizing the dynamic prior $p_f(\boldsymbol{z}|\boldsymbol{X})$, the sum of KL-divergences in Equation (6) can be written as JS-divergence (see Equation (4)) if the function $f$ defines a mixture distribution. To remain a valid ELBO, the function $p_f(\boldsymbol{z}|\boldsymbol{X})$ needs to be a well-defined prior.

**Lemma 1.** *If the function $f$ of the dynamic prior $p_f(\boldsymbol{z}|\boldsymbol{X})$ defines a mixture distribution of the unimodal approximation distributions $\{q_{\phi_\nu}(\boldsymbol{z}|\boldsymbol{x}_\nu)\}_{\nu=1}^M$, the resulting dynamic prior $p_{MoE}(\boldsymbol{z}|\boldsymbol{X})$ is well-defined.*

*Proof.* The proof can be found in Appendix B. $\qquad\square$

With Lemma 1, the new multimodal objective $\widetilde{\mathcal{L}}(\theta, \phi; \boldsymbol{X})$ utilizing the Jensen-Shannon divergence (Definition 1) can now be directly derived from the ELBO in Equation (2).

**Lemma 2.** *The multimodal objective $\widetilde{\mathcal{L}}(\theta, \phi; \boldsymbol{X})$ utilizing the Jensen-Shannon divergence defined in Equation (5) is a lower bound to the ELBO in Equation (2).*

$$\mathcal{L}(\theta, \phi; \boldsymbol{X}) \geq \widetilde{\mathcal{L}}(\theta, \phi; \boldsymbol{X}) \tag{8}$$

*Proof.* The lower bound to the ELBO in Equation (6) can be rewritten using the dynamic prior $p_{\text{MoE}}(\boldsymbol{z}|\boldsymbol{X})$:

$$
\begin{aligned}
\mathcal{L}(\theta, \phi; \boldsymbol{X}) \geq &E_{q_\phi(\boldsymbol{z}|\boldsymbol{X})}[\log p_\theta(\boldsymbol{X}|\boldsymbol{z})] - \sum_{j=1}^M \pi_j KL(q_{\phi_j}(\boldsymbol{z}|\boldsymbol{x}_j)||p_{\text{MoE}}(\boldsymbol{z}|\boldsymbol{X})) \\
&- \pi_{M+1} KL(p_\theta(\boldsymbol{z})||p_{\text{MoE}}(\boldsymbol{z}|\boldsymbol{X})) \\
= &E_{q_\phi(\boldsymbol{z}|\boldsymbol{X})}[\log p_\theta(\boldsymbol{X}|\boldsymbol{z})] - JS_{\boldsymbol{\pi}}^{M+1}(\{q_{\phi_j}(\boldsymbol{z}|\boldsymbol{x}_j)\}_{j=1}^M, p_\theta(\boldsymbol{z})) \\
= &\widetilde{\mathcal{L}}(\theta, \phi; \boldsymbol{X})
\end{aligned}
\tag{9}
$$

Proving that $\widetilde{\mathcal{L}}(\theta, \phi; \boldsymbol{X})$ is a lower bound to the original ELBO formulation in Equation (2) also proves that it is a lower bound the marginal log-likelihood $\log p_\theta(\boldsymbol{X}^{(i)})$. This makes the proposed objective an ELBO itself.[1] $\qquad\square$

The objective in Equation (5) using the JS-divergence is an intuitive extension of the ELBO formulation to the multimodal case as it relates the unimodal to the multimodal approximation functions while providing a more expressive prior [25]. In addition, it is important to notice that the function $f$ of the dynamic prior $p_f(\boldsymbol{z}|\boldsymbol{X})$, e.g. an arithmetic mean as in $p_{\text{MoE}}(\boldsymbol{z}|\boldsymbol{X})$, is not related to the definition of the joint posterior approximation $q_\phi(\boldsymbol{z}|\boldsymbol{X})$. Hence, Definition 1 is a special case which follows the definition of the dynamic prior $p_f(\boldsymbol{z}|\boldsymbol{X})$ as $p_{\text{MoE}}(\boldsymbol{z}|\boldsymbol{X})$ – or other abstract mean functions (see Section 3.4).

### 3.4  Generalized Jensen-Shannon Divergence

[19] defines the JS-divergence for the general case of abstract means. This allows to calculate the JS-divergence not only using an arithmetic mean as in the standard formulation, but any mean function. Abstract means are a suitable class of functions for aggregating information from different distributions while being able to handle missing data [19].

**Definition 3.** *The dynamic prior $p_{PoE}(\boldsymbol{z}|\boldsymbol{X})$ is defined as the geometric mean of the unimodal posterior approximations $\{q_{\phi_\nu}(\boldsymbol{z}|\boldsymbol{x}_\nu)\}_{\nu=1}^M$ and the pre-defined distribution $p_\theta(\boldsymbol{z})$.*

For Gaussian distributed arguments, the geometric mean is again Gaussian distributed and equivalent to a weighted PoE [8]. The proof that $p_{\text{PoE}}(\boldsymbol{z}|\boldsymbol{X})$ is a well-defined prior can be found in Appendix B.

Utilizing Definition 3, the JS-divergence in Equation (5) can be calculated in closed form. This allows the optimization of the proposed, multimodal objective $\widetilde{\mathcal{L}}(\theta, \phi; \boldsymbol{X})$ in a computationally efficient way. As the unimodal posterior approximations are directly optimized as well, $\widetilde{\mathcal{L}}(\theta, \phi; \boldsymbol{X})$ using a PoE-prior also tackles the limitations of previous work outlined in Section 2.1. Hence, we use a dynamic prior of the form $p_{\text{PoE}}(\boldsymbol{z}|\boldsymbol{X})$ for our experiments.

### 3.5  Modality-specific Latent Subspaces

We define our latent representations as a combination of modality-specific spaces and a shared, modality-independent space: $\boldsymbol{z} = (\boldsymbol{S}, \boldsymbol{c}) = (\{\boldsymbol{s}_j\}_{j=1}^M, \boldsymbol{c})$. Every $\boldsymbol{x}_j$ is modelled to have its own independent, modality-specific part $\boldsymbol{s}_j$. Additionally, we assume a joint content $\boldsymbol{c}$ for all $\boldsymbol{x}_j \in \boldsymbol{X}$ which captures the information that is shared across modalities. $\boldsymbol{S}$ and $\boldsymbol{c}$ are considered conditionally independent given $\boldsymbol{X}$. Different to previous work [3, 26, 28], we empirically show that meaningful representations can be learned in a self-supervised setting by the supervision which is given naturally for multimodal problems. Building on what we derived in Sections 2 and 3, and the assumptions outlined above, we model the modality-dependent divergence term similarly to the unimodal setting as there is no intermodality relationship associated with them. Applying these assumptions to Equation (5), it follows (for details, please see Appendix B):

$$\widetilde{\mathcal{L}}(\theta, \phi; \boldsymbol{X}) = \sum_{j=1}^M E_{q_{\phi_{\boldsymbol{c}}}(\boldsymbol{c}|\boldsymbol{X})}[E_{q_{\phi_{\boldsymbol{s}_j}}(\boldsymbol{s}_j|\boldsymbol{x}_j)}[\log p_\theta(\boldsymbol{x}_j|\boldsymbol{s}_j, \boldsymbol{c})]] \tag{10}$$
$$- \sum_{j=1}^M D_{KL}(q_{\phi_{\boldsymbol{s}_j}}(\boldsymbol{s}_j|\boldsymbol{x}_j)||p_\theta(\boldsymbol{s}_j)) - JS_{\boldsymbol{\pi}}^{M+1}(\{q_{\phi_{\boldsymbol{c}_j}}(\boldsymbol{c}|\boldsymbol{x}_j)\}_{j=1}^M, p_\theta(\boldsymbol{c}))$$

The objective in Equation (5) is split further into two different divergence terms: The JS-divergence is used only for the multimodal latent factors $\boldsymbol{c}$, while modality-independent terms $\boldsymbol{s}_j$ are part of a sum of KL-divergences. Following the common line in VAE-research, the variational approximation functions $q_{\phi_{\boldsymbol{c}_j}}(\boldsymbol{c}_j|\boldsymbol{x}_j)$ and $q_{\phi_{\boldsymbol{s}_j}}(\boldsymbol{s}_j|\boldsymbol{x}_j)$, as well as the generative models $p_\theta(\boldsymbol{x}_j|\boldsymbol{s}_j, \boldsymbol{c})$ are parameterized by neural networks.

## 4  Experiments & Results

We carry out experiments on two different datasets[2]. For the experiment we use a matching digits dataset consisting of MNIST [14] and SVHN [17] images with an additional text modality [22]. This experiment provides empirical evidence on a method's generalizability to more than two modalities.

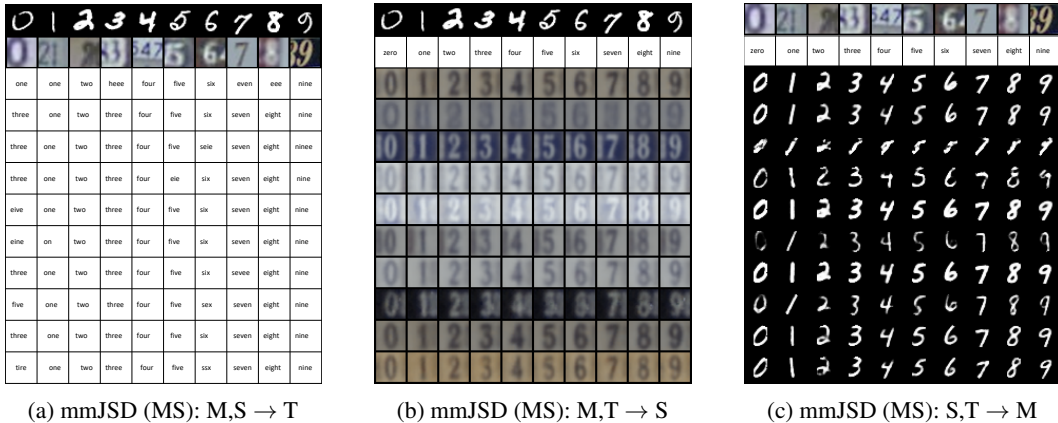

| (a) mmJSD (MS): M,S → T | (b) mmJSD (MS): M,T → S | (c) mmJSD (MS): S,T → M |

Figure 1: Qualitative results for missing data estimation. Each row is generated by a single, random style and the information inferred from the available modalities in the first two rows. This allows for the generation of samples with coherent, random styles across multiple contents (see Table 1 for explanation of abbreviations).

The second experiment is carried out on the challenging CelebA faces dataset [16] with additional text describing the attributes of the shown face. The CelebA dataset is highly imbalanced regarding the distribution of attributes which poses additional challenges for generative models.

## 4.1 Evaluation

We evaluate the performance of models with respect to the multimodal wish-list introduced in Section 1. To assess the discriminative capabilities of a model, we evaluate the latent representations with respect to the input data's semantic information. We employ a linear classifier on the unimodal and multimodal posterior approximations. To assess the generative performance, we evaluate generated samples according to their quality and coherence [22]. Generation should be coherent across all modalities with respect to shared information. Conditionally generated samples should be coherent with the input data, randomly generated samples with each other. To evaluate the coherence of generated samples, we use a classifier which was trained on the original unimodal training set. If the classifier detects the same attributes in the samples, it is a coherent generation [20]. To assess the quality of generated samples, we use the precision and recall metric for generative models [21] where precision defines the quality and recall the diversity of the generated samples. In addition, we evaluate all models regarding their test set log-likelihoods.

We compare the proposed method to two state-of-the-art models: the MVAE model [29] and the MMVAE model [22] described in Section 2.1. We use the same encoder and decoder networks and the same number of parameters for all methods. Implementation details for all experiments together with a comparison of runtimes can be found in Appendix C.

## 4.2 MNIST-SVHN-Text

Previous works on scalable, multimodal methods performed no evaluation on more than two modalities[3]. We use the MNIST-SVHN dataset [22] as basis. To this dataset, we add an additional, text-based modality. The texts consist of strings which name the digit in English where the start index of the word is chosen at random to have more diversity in the data. To evaluate the effect of the dynamic prior as well as modality-specific latent subspaces, we first compare models with a single shared latent space. In a second comparison, we add modality-specific subspaces to all models (for these experiments, we add a (MS)-suffix to the model names). This allows us to assess and evaluate the contribution of the dynamic prior as well as modality-specific subspaces. Different subspace sizes are compared in Appendix C.

Table 1: Classification accuracy of the learned latent representations using a linear classifier. We evaluate all subsets of modalities for which we use the following abbreviations: M: MNIST; S: SVHN; T: Text; M,S: MNIST and SVHN; M,T: MNIST and Text; S,T: SVHN and Text; Joint: all modalities. (MS) names the models with modality-specific latent subspaces.

| MODEL | M | S | T | M,S | M,T | S,T | JOINT |
|---|---|---|---|---|---|---|---|
| MVAE | 0.85 | 0.20 | 0.58 | 0.80 | 0.92 | 0.46 | 0.90 |
| MMVAE | 0.96 | 0.81 | **0.99** | 0.89 | 0.97 | 0.90 | 0.93 |
| MMJSD | 0.97 | 0.82 | **0.99** | 0.93 | **0.99** | 0.92 | 0.98 |
| MVAE (MS) | 0.86 | 0.28 | 0.78 | 0.82 | 0.94 | 0.64 | 0.92 |
| MMVAE (MS) | 0.96 | 0.81 | **0.99** | 0.89 | 0.98 | 0.91 | 0.92 |
| MMJSD (MS) | **0.98** | **0.85** | 0.99 | **0.94** | 0.98 | **0.94** | **0.99** |

Table 2: Classification accuracy of generated samples on MNIST-SVHN-Text. In case of conditional generation, the letter above the horizontal line indicates the modality which is generated based on the different sets of modalities below the horizontal line.

| MODEL | RANDOM | M | | | S | | | T | | |
|---|---|---|---|---|---|---|---|---|---|---|
| | | S | T | S,T | M | T | M,T | M | S | M,S |
| MVAE | 0.72 | 0.17 | 0.14 | 0.22 | 0.37 | 0.30 | 0.86 | 0.20 | 0.12 | 0.22 |
| MMVAE | 0.54 | **0.82** | **0.99** | 0.91 | 0.32 | 0.30 | 0.31 | 0.96 | **0.83** | 0.90 |
| MMJSD | 0.60 | **0.82** | **0.99** | **0.95** | 0.37 | 0.36 | 0.48 | **0.97** | **0.83** | **0.92** |
| MVAE (MS) | **0.74** | 0.16 | 0.17 | 0.25 | 0.35 | 0.37 | 0.85 | 0.24 | 0.14 | 0.26 |
| MMVAE (MS) | 0.67 | 0.77 | 0.97 | 0.86 | 0.88 | **0.93** | 0.90 | 0.82 | 0.70 | 0.76 |
| MMJSD (MS) | 0.66 | 0.80 | 0.97 | 0.93 | **0.89** | **0.93** | **0.92** | 0.92 | 0.79 | 0.86 |

Tables 1 and 2 demonstrate that the proposed mmJSD objective generalizes better to three modalities than previous work. The difficulty of the MVAE objective in optimizing the unimodal posterior approximation is reflected in the coherence numbers of missing data types and the latent representation classification. Although MMVAE is able to produce good results if only a single data type is given, the model cannot leverage the additional information of multiple available observations. Given multiple modalities, the corresponding performance numbers are the arithmetic mean of their unimodal pendants. The mmJSD model is able to achieve state-of-the-art performance in optimizing the unimodal posteriors as well as outperforming previous work in leveraging multiple modalities thanks to the dynamic prior. The quality of random samples might be affected by the dynamic prior: this needs to be investigated further in future work. The introduction of modality-specific subspaces increases the coherence of the difficult SVHN modality for MMVAE and mmJSD. More importantly, modality-specific latent spaces improve the quality of the generated samples for all modalities (see Table 3). Figure 1 shows qualitative results. Table 4 provides evidence that the high coherence of generated samples of the mmJSD model are not traded off against test set log-likelihoods. It also shows that MVAE is able to learn the statistics of a dataset well, but not to preserve the content in case of missing modalities.

## 4.3   Bimodal CelebA

Every CelebA image is labelled according to 40 attributes. We extend the dataset with an additional text modality describing the face in the image using the labelled attributes. Examples of created strings can be seen in Figure 2. Any negative attribute is completely absent in the string. This is different and more difficult to learn than negated attributes as there is no fixed position for a certain attribute in a string which introduces additional variability in the data. Figure 2 shows qualitative results for images which are generated conditioned on text samples. Every row of images is based on the text next to it. As the labelled attributes are not capturing all possible variation of a face, we generate 10 images with randomly sampled image-specific information to capture the distribution of information which is not encoded in the shared latent space. The imbalance of some attributes affects

Table 3: Quality of generated samples on MNIST-SVHN-Text. We report the average precision based on the precision-recall metric for generative models (higher is better) for conditionally and randomly generated image data (R: Random Generation).

| | M | | | | S | | | |
|---|---|---|---|---|---|---|---|---|
| MODEL | S | T | S,T | R | M | T | M, T | R |
| MVAE | **0.62** | 0.62 | 0.58 | **0.62** | **0.33** | **0.34** | 0.22 | **0.33** |
| MMVAE | 0.22 | 0.09 | 0.18 | 0.35 | 0.005 | 0.006 | 0.006 | 0.27 |
| MMJSD | 0.19 | 0.09 | 0.16 | 0.15 | 0.05 | 0.01 | 0.06 | 0.09 |
| MVAE (MS) | 0.60 | 0.59 | 0.50 | 0.60 | 0.30 | 0.33 | 0.17 | 0.29 |
| MMVAE (MS) | **0.62** | 0.63 | 0.63 | 0.52 | 0.21 | 0.20 | 0.20 | 0.19 |
| MMJSD (MS) | **0.62** | **0.64** | **0.64** | 0.30 | 0.21 | 0.22 | **0.22** | 0.17 |

Table 4: Test set log-likelihoods on MNIST-SVHN-Text. We report the log-likelihood of the joint generative model $p_\theta(\boldsymbol{X})$ and the log-likelihoods of the joint generative model conditioned on the variational posterior of subsets of modalities $q_{\phi_K}(\boldsymbol{z}|\boldsymbol{X}_K)$. ($\boldsymbol{x}_M$: MNIST; $\boldsymbol{x}_S$: SVHN; $\boldsymbol{x}_T$: Text; $\boldsymbol{X} = (\boldsymbol{x}_M, \boldsymbol{x}_S, \boldsymbol{x}_T)$).

| MODEL | $\boldsymbol{X}$ | $\boldsymbol{X}|\boldsymbol{x}_M$ | $\boldsymbol{X}|\boldsymbol{x}_S$ | $\boldsymbol{X}|\boldsymbol{x}_T$ | $\boldsymbol{X}|\boldsymbol{x}_M,\boldsymbol{x}_S$ | $\boldsymbol{X}|\boldsymbol{x}_M,\boldsymbol{x}_T$ | $\boldsymbol{X}|\boldsymbol{x}_S,\boldsymbol{x}_T$ |
|---|---|---|---|---|---|---|---|
| MVAE | **-1864** | -2002 | -1936 | -2040 | **-1881** | -1970 | **-1908** |
| MMVAE | -1916 | -2213 | **-1911** | -2250 | -2062 | -2231 | -2080 |
| MMJSD | -1961 | -2175 | -1955 | -2249 | -2000 | -2121 | -2004 |
| MVAE (MS) | -1870 | -1999 | -1937 | -2033 | -1886 | -1971 | -1909 |
| MMVAE (MS) | -1893 | **-1982** | -1934 | **-1995** | -1905 | **-1958** | -1915 |
| MMJSD (MS) | -1900 | -1994 | -1944 | -2006 | -1907 | -1968 | -1918 |

the generative process. Rare and subtle attributes like eyeglasses are difficult to learn while frequent attributes like gender and smiling are well learnt.

Table 5 demonstrates the superior performance of the proposed mmJSD objective compared to previous work on the challening bimodal CelebA dataset. The classification results regarding the individual attributes can be found in Appendix C.

Table 5: Classfication results on the bimodal CelebA experiment. For latent representations and conditionally generated samples, we report the mean average precision over all attributes (I: Image; T: Text; Joint: I and T).

| | LATENT REPRESENTATION | | | GENERATION | |
|---|---|---|---|---|---|
| MODEL | I | T | JOINT | I $\rightarrow$ T | T $\rightarrow$ I |
| MVAE (MS) | 0.42 | 0.45 | 0.44 | **0.32** | 0.30 |
| MMVAE (MS) | 0.43 | 0.45 | 0.42 | 0.30 | 0.36 |
| MMJSD (MS) | **0.48** | **0.59** | **0.57** | **0.32** | **0.42** |

## 5  Conclusion

In this work, we propose a novel generative model for learning from multimodal data. Our contributions are fourfold: (i) we formulate a new multimodal objective using a dynamic prior. (ii) We propose to use the JS-divergence for multiple distributions as a divergence measure for multimodal data. This measure enables direct optimization of the unimodal as well as the joint latent approximation functions. (iii) We prove that the proposed mmJSD objective constitutes an ELBO for multiple data types. (iv) With the introduction of modality-specific latent spaces, we show empirically the improvement in quality of generated samples. Additionally, we demonstrate that the proposed method does not need any additional training objectives while reaching state-of-the-art or superior

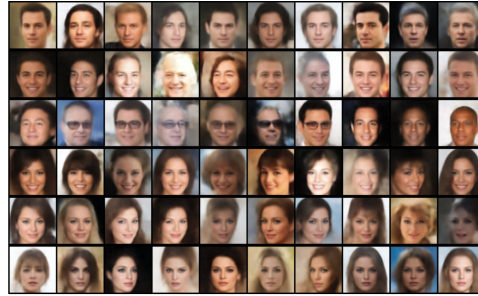

5 o clock shadow, attractive, black hair, male, straight hair, young

attractive, male, mouth slightly open, no beard, smiling, wearing hat, young

eyeglasses, high cheekbones, male, mouth slightly open, no beard, smiling, straight hair, wearing necktie

attractive, bangs, black hair, heavy makeup, high cheekbones, mouth slightly open, no beard, oval face, pointy nose, smiling, wearing lipstick, young

attractive, big lips, blond hair, heavy makeup, high cheekbones, no beard, oval face, smiling, wavy hair, wearing lipstick, wearing necklace, young

big lips, black hair, no beard, receding hairline, young

Figure 2: Qualitative Results of CelebA faces which were conditionally generated based on text strings using mmJSD.

performance compared to recently proposed, scalable, multimodal generative models. In future work, we would like to further investigate which functions $f$ would serve well as prior function and we will apply our proposed model in the medical domain.

## 6   Broader Impact

Learning from multiple data types offers many potential applications and opportunities as multiple data types naturally co-occur. We intend to apply our model in the medical domain in future work, and we will focus here on the impact our model might have in the medical application area. Models that are capable of dealing with large-scale multi-modal data are extremely important in the field of computational medicine and clinical data analysis. The recent developments in medical information technology have resulted in an overwhelming amount of multi-modal data available for every single patient. A patient visit at a hospital may result in tens of thousands of measurements and structured information, including clinical factors, diagnostic imaging, lab tests, genomic and proteomic tests, and hospitals may see thousands of patients each year. The ultimate aim is to use all this vast information for a medical treatment tailored to the needs of an individual patient. To turn the vision of precision medicine into reality, there is an urgent need for the integration of the multi-modal patient data currently available for improved disease diagnosis, prognosis and therapy outcome prediction. Instead of learning on one data set exclusively, as for example just on images or just on genetics, the aim is to improve learning and enhance personalized treatment by using as much information as possible for every patient. First steps in this direction have been successful, but so far a major hurdle has been the huge amount of heterogeneous data with many missing data points which is collected for every patient.

With this work, we lay the theoretical foundation for the analysis of large-scale multi-modal data. We focus on a self-supervised approach as collecting labels for large datasets of multiple data types is expensive and becomes quickly infeasible with a growing number of modalities. Self-supervised approaches have the potential to overcome the need for excessive labelling and the bias coming from these labels. In this work, we extensively tested the model in controlled environments. In future work, we will apply our proposed model to medical multi-modal data with the goal of gaining insights and making predictions about disease phenotypes, disease progression and response to treatment.

### Acknowledgments and Disclosure of Funding

Thanks to Diane Bouchacourt for providing code and Ričards Marcinkevičs for helpful discussions. ID is supported by the SNSF grant #200021_188466.

## Footnotes

[1] We would like to emphasize that the lower bound in the first line of Equation (9) originates from Equation (6) and not from the introduction of the dynamic prior.

[2]The code for our experiments can be found here.

[3][29] designed a computer vision study with multiple transformations and a multimodal experiment for the CelebA dataset where every attribute is considered a modality.

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
