[Supplementary Material]

# Supplementary Material: Multimodal Generative Learning Utilizing Jensen-Shannon-Divergence

**Thomas M. Sutter,   Imant Daunhawer,   Julia E. Vogt**
Department of Computer Science
ETH Zurich
{thomas.sutter,imant.daunhawer,julia.vogt}@inf.ethz.ch

In the supplementary material section, we provide additional mathematical derivations, implementation details and results which could not be put in the main paper due to space restrictions.

## A   Theoretical Background

The ELBO $\mathcal{L}(\theta, \phi; \boldsymbol{X})$ can be derived by reformulating the KL-divergence between the joint posterior approximation function $q_\phi(\boldsymbol{z}|\boldsymbol{X})$ and the true posterior distribution $p_\theta(\boldsymbol{z}|\boldsymbol{X})$:

$$
\begin{aligned}
KL(q_\phi(\boldsymbol{z}|\boldsymbol{X})||p_\theta(\boldsymbol{z}|\boldsymbol{X})) &= \int_{\boldsymbol{z}} q_\phi(\boldsymbol{z}|\boldsymbol{X}) \log\left(\frac{q_\phi(\boldsymbol{z}|\boldsymbol{X})}{p_\theta(\boldsymbol{z}|\boldsymbol{X})}\right) dz \\
&= \int_{\boldsymbol{z}} q_\phi(\boldsymbol{z}|\boldsymbol{X}) \log\left(\frac{q_\phi(\boldsymbol{z}|\boldsymbol{X})p_\theta(\boldsymbol{X})}{p_\theta(\boldsymbol{X},\boldsymbol{z})}\right) dz \\
&= E_{q_\phi}[\log(q_\phi(\boldsymbol{z}|\boldsymbol{X})) - \log(p_\theta(\boldsymbol{X},\boldsymbol{z}))] + \log(p_\theta(\boldsymbol{X}))
\end{aligned}
\tag{1}
$$

It follows:

$$
\log p_\theta(\boldsymbol{X}) = KL(q_\phi(\boldsymbol{z}|\boldsymbol{X})||p_\theta(\boldsymbol{z}|\boldsymbol{X})) - E_{q_\phi}[\log(q_\phi(\boldsymbol{z}|\boldsymbol{X})) - \log(p_\theta(\boldsymbol{X},\boldsymbol{z}))]
\tag{2}
$$

From the non-negativity of the KL-divergence, it directly follows:

$$
\mathcal{L}(\theta, \phi; \boldsymbol{X}) = E_{q_\phi(\boldsymbol{z}|\boldsymbol{X})}[\log(p_\theta(\boldsymbol{X}|\boldsymbol{z})] - KL(q_\phi(\boldsymbol{z}|\boldsymbol{X})||p_\theta(\boldsymbol{z}))
\tag{3}
$$

In the absence of one or multiple data types, we would still like to be able to approximate the true multimodal posterior distribution $p_\theta(\boldsymbol{z}|\boldsymbol{X})$. However, we are only able to approximate the posterior by a variational function $q_\phi(\boldsymbol{z}|\boldsymbol{X}_K)$ with $K \leq M$. In addition, for different samples, different modalities might be missing. The derivation of the ELBO formulation changes accordingly:

$$
\begin{aligned}
KL(q_{\phi_K}(\boldsymbol{z}|\boldsymbol{X}_K)||p_\theta(\boldsymbol{z}|\boldsymbol{X})) &= \int_{\boldsymbol{z}} q_\phi(\boldsymbol{z}|\boldsymbol{X}_K) \log\left(\frac{q_\phi(\boldsymbol{z}|\boldsymbol{X}_K)}{p_\theta(\boldsymbol{z}|\boldsymbol{X})}\right) dz \\
&= \int_{\boldsymbol{z}} q_\phi(\boldsymbol{z}|\boldsymbol{X}_K) \log\left(\frac{q_\phi(\boldsymbol{z}|\boldsymbol{X}_K)p_\theta(\boldsymbol{X})}{p_\theta(\boldsymbol{X},\boldsymbol{z})}\right) dz \\
&= E_{q_\phi}[\log(q_\phi(\boldsymbol{z}|\boldsymbol{X}_K)) - \log(p_\theta(\boldsymbol{X},\boldsymbol{z}))] + \log(p_\theta(\boldsymbol{X}))
\end{aligned}
\tag{4}
$$

From where it again follows:

$$
\mathcal{L}_K(\theta, \phi_K, \boldsymbol{X}) = E_{q_\phi(\boldsymbol{z}|\boldsymbol{X}_K)}[\log(p_\theta(\boldsymbol{X}|\boldsymbol{z})] - KL(q_\phi(\boldsymbol{z}|\boldsymbol{X}_K)||p_\theta(\boldsymbol{z}))
\tag{5}
$$

## B   Multimodal Jensen-Shannon Divergence Objective

In this section, we provide the proofs to the Lemmas which were introduced in the main paper. Due to space restrictions, the proofs of these Lemmas had to be moved to the appendix.

### B.1 Upper bound to the KL-divergence of a mixture distribution

**Lemma 1** (Joint Approximation Function). *Under the assumption of $q_\phi(\boldsymbol{z}|\{\boldsymbol{x}_j\}_{j=1}^M)$ being a mixture model of the unimodal variational posterior approximations $q_{\phi_j}(\boldsymbol{z}|\boldsymbol{x}_j)$, the KL-divergence of the multimodal variational posterior approximation $q_\phi(\boldsymbol{z}|\{\boldsymbol{x}_j\}_{j=1}^M)$ is a lower bound for the weighted sum of the KL-divergences of the unimodal variational approximation functions $q_{\phi_j}(\boldsymbol{z}|\boldsymbol{x}_j)$:*

$$KL(\sum_{j=1}^M \pi_j q_{\phi_j}(\boldsymbol{z}|\boldsymbol{x}_j)||p_\theta(\boldsymbol{z})) \leq \sum_{j=1}^M \pi_j KL(q_{\phi_j}(\boldsymbol{z}|\boldsymbol{x}_j)||p_\theta(\boldsymbol{z})) \tag{6}$$

*Proof.* Lemma 1 follows directly from the strict convexity of $g(t) = t \log t$. □

### B.2 MoE-Prior

**Definition 1** (MoE-Prior). *The prior $p_{MoE}(\boldsymbol{z}|\boldsymbol{X})$ is defined as follows:*

$$p_{MoE}(\boldsymbol{z}|\boldsymbol{X}) = \sum_{\nu=1}^M \pi_\nu q_{\phi_\nu}(\boldsymbol{z}|\boldsymbol{x}_\nu) + \pi_{M+1}p_\theta(\boldsymbol{z}) \tag{7}$$

*where $q_{\phi_\nu}(\boldsymbol{z}|\boldsymbol{x}_\nu)$ are again the unimodal approximation functions and $p_\theta(\boldsymbol{z})$ is a pre-defined, parameterizable distribution. The mixture weights $\boldsymbol{\pi}$ sum to one, i.e. $\sum \pi_j = 1$.*

We prove that the MoE-prior $p_{MoE}(\boldsymbol{z}|\boldsymbol{X})$ is a well-defined prior:

**Lemma 2.** *If the function $f$ of the dynamic prior $p_f(\boldsymbol{z}|\boldsymbol{X})$ defines a mixture distribution of the unimodal approximation distributions $q_{\phi_\nu}(\boldsymbol{z}|\boldsymbol{x}_\nu)$, the resulting dynamic prior is well-defined.*

*Proof.* To be a well-defined prior, $p_{MoE}(\boldsymbol{z}|\boldsymbol{X})$ must satisfy the following condition:

$$\int p_{MoE}(\boldsymbol{z}|\boldsymbol{X})d\boldsymbol{z} = 1 \tag{8}$$

Therefore,

$$\int \left( \sum_{\nu=1}^M \pi_\nu q_{\phi_\nu}(\boldsymbol{z}|\boldsymbol{x}_\nu) + \pi_{M+1}p_\theta(\boldsymbol{z}) \right) d\boldsymbol{z}$$

$$= \sum_{\nu=1}^M \pi_\nu \int q_{\phi_\nu}(\boldsymbol{z}|\boldsymbol{x}_\nu)d\boldsymbol{z} + \pi_{M+1} \int p_\theta(\boldsymbol{z})d\boldsymbol{z}$$

$$= \sum_{\nu=1}^M \pi_\nu + \pi_{M+1} = 1 \tag{9}$$

The unimodal approximation functions $q_{\phi_\nu}(\boldsymbol{z}|\boldsymbol{x}_\nu)$ as well as the pre-defined distribution $p_\theta(\boldsymbol{z})$ are well-defined probability distributions. Hence, $\int q_{\phi_\nu}(\boldsymbol{z}|\boldsymbol{x}_\nu)d\boldsymbol{z} = 1$ for all $q_{\phi_\nu}(\boldsymbol{z}|\boldsymbol{x}_\nu)$ and $\int p_\theta(\boldsymbol{z})d\boldsymbol{z} = 1$. The last line in equation 9 follows from the assumptions. Therefore, equation (7) is a well-defined prior. □

### B.3 PoE-Prior

**Lemma 3.** *Under the assumption that all $q_{\phi_\nu}(\boldsymbol{z}|\boldsymbol{x}_\nu)$ are Gaussian distributed by $\mathcal{N}(\boldsymbol{\mu}_\nu(\boldsymbol{x}_\nu), \boldsymbol{\sigma}_\nu^2(\boldsymbol{x}_\nu)\boldsymbol{I})$, $p_{PoE}(\boldsymbol{z}|\boldsymbol{X})$ is Gaussian distributed:*

$$p_{PoE}(\boldsymbol{z}|\boldsymbol{X}) \sim \mathcal{N}(\boldsymbol{\mu}_{GM}, \boldsymbol{\sigma}_{GM}^2\boldsymbol{I}) \tag{10}$$

*where $\boldsymbol{\mu}_{GM}$ and $\boldsymbol{\sigma}_{GM}^2\boldsymbol{I}$ are defined as follows:*

$$\boldsymbol{\sigma}_{GM}^2\boldsymbol{I} = (\sum_{k=1}^{M+1} \pi_k \boldsymbol{\sigma}_k^2\boldsymbol{I})^{-1}, \qquad \boldsymbol{\mu}_{GM} = (\boldsymbol{\sigma}_{GM}^2\boldsymbol{I}) \sum_{k=1}^{M+1} \pi_k(\boldsymbol{\sigma}_k^2\boldsymbol{I})^{-1}\boldsymbol{\mu}_k \tag{11}$$

*which makes $p_{PoE}(\boldsymbol{z}|\boldsymbol{X})$ a well-defined prior.*

*Proof.* As $p_{PoE}(\boldsymbol{z}|\boldsymbol{X})$ is Gaussian distributed, it follows immediately that $p_{PoE}(\boldsymbol{z}|\boldsymbol{X})$ is a well-defined dynamic prior. □

## B.4 Factorization of Representations

We mostly base our derivation of factorized representations on the paper by Bouchacourt et al. [1]. Tsai et al. [8] and Hsu and Glass [4] used a similar idea. A set $\boldsymbol{X}$ of modalities can be seen as group and analogous every modality as a member of a group. We model every $\boldsymbol{x}_j$ to have its own modality-specific latent code $\boldsymbol{s}_j \in \boldsymbol{S}$.

$$\boldsymbol{S} = (\boldsymbol{s}_j, \forall \boldsymbol{x}_j \in \boldsymbol{X}) \tag{12}$$

From Equation (12), we see that $\boldsymbol{S}$ is the collection of all modality-specific latent variables for the set $\boldsymbol{X}$. Contrary to this, the modality-invariant latent code $\boldsymbol{c}$ is shared between all modalities $\boldsymbol{x}_j$ of the set $\boldsymbol{X}$. Also like Bouchacourt et al. [1], we model the variational approximation function $q_\phi(\boldsymbol{S}, \boldsymbol{c})$ to be conditionally independent given $\boldsymbol{X}$, i.e.:

$$q_\phi(\boldsymbol{S}, \boldsymbol{c}) = q_{\phi_{\boldsymbol{S}}}(\boldsymbol{S}|\boldsymbol{X}) q_{\phi_{\boldsymbol{c}}}(\boldsymbol{c}|\boldsymbol{X}) \tag{13}$$

From the assumptions it is clear that $q_{\phi_{\boldsymbol{S}}}$ factorizes:

$$q_{\phi_{\boldsymbol{S}}}(\boldsymbol{S}|\boldsymbol{X}) = \prod_{j=1}^{M} q_{\phi_{\boldsymbol{s}_j}}(\boldsymbol{s}_j|\boldsymbol{x}_j) \tag{14}$$

From Equation (14) and the fact that the multimodal relationships are only modelled by the latent factor $\boldsymbol{c}$, it is reasonable to only apply the mmJSD objective to $\boldsymbol{c}$. It follows:

$$
\begin{aligned}
\mathcal{L}(\theta, \phi; \boldsymbol{X}) =& E_{q_\phi(\boldsymbol{z}|\boldsymbol{X})}[\log p_\theta(\boldsymbol{X}|\boldsymbol{z})] - KL(q_\phi(\boldsymbol{z}|\boldsymbol{X})\|p_\theta(\boldsymbol{z})) \\
=& E_{q_\phi(\boldsymbol{S}, \boldsymbol{c}|\boldsymbol{X})}[\log p_\theta(\boldsymbol{X}|\boldsymbol{S}, \boldsymbol{c})] - KL(q_\phi(\boldsymbol{S}, \boldsymbol{c}|\boldsymbol{X})\|p_\theta(\boldsymbol{S}, \boldsymbol{c})) \\
=& E_{q_\phi(\boldsymbol{S}, \boldsymbol{c}|\boldsymbol{X})}[\log p_\theta(\boldsymbol{X}|\boldsymbol{S}, \boldsymbol{c})] - KL(q_{\phi_{\boldsymbol{S}}}(\boldsymbol{S}|\boldsymbol{X})\|p_\theta(\boldsymbol{S})) - KL(q_{\phi_{\boldsymbol{c}}}(\boldsymbol{c}|\boldsymbol{X})\|p_f(\boldsymbol{c})) \\
=& E_{q_\phi(\boldsymbol{S}, \boldsymbol{c}|\boldsymbol{X})}[\log p_\theta(\boldsymbol{X}|\boldsymbol{S}, \boldsymbol{c})] - \sum_{j=1}^{M} KL(q_{\phi_{\boldsymbol{s}_j}}(\boldsymbol{s}_j|\boldsymbol{x}_j)\|p_\theta(\boldsymbol{s}_j)) - KL(q_{\phi_{\boldsymbol{c}}}(\boldsymbol{c}|\boldsymbol{X})\|p_f(\boldsymbol{c}))
\end{aligned}
\tag{15}
$$

In Equation (15), we can rewrite the KL-divergence which includes $\boldsymbol{c}$ using the multimodal dynamic prior and the JS-divergence for multiple distributions:

$$
\begin{aligned}
\widetilde{\mathcal{L}}(\theta, \phi; \boldsymbol{X}) =& E_{q_\phi(\boldsymbol{S}, \boldsymbol{c}|\boldsymbol{X})}[\log p_\theta(\boldsymbol{X}|\boldsymbol{S}, \boldsymbol{c})] - \sum_{j=1}^{M} KL(q_{\phi_{\boldsymbol{s}_j}}(\boldsymbol{s}_j|\boldsymbol{x}_j)\|p_\theta(\boldsymbol{s}_j)) \\
& - JS_{\boldsymbol{\pi}}^{M+1}(\{q_{\phi_{\boldsymbol{c}_j}}(\boldsymbol{c}|\boldsymbol{x}_j)\}_{j=1}^{M}, p_\theta(\boldsymbol{c}))
\end{aligned}
\tag{16}
$$

The expectation over $q_\phi(\boldsymbol{S}, \boldsymbol{c}|\boldsymbol{X})$ can be rewritten as a concatenation of expectations over $q_{\phi_{\boldsymbol{c}}}(\boldsymbol{c}|\boldsymbol{X})$ and $q_{\phi_{\boldsymbol{s}_j}}(\boldsymbol{s}_j|\boldsymbol{x}_j)$:

$$
\begin{aligned}
E_{q_\phi(\boldsymbol{S}, \boldsymbol{c}|\boldsymbol{X})}[\log p_\theta(\boldsymbol{X}|\boldsymbol{S}, \boldsymbol{c})] =& \int_{\boldsymbol{c}} \int_{\boldsymbol{S}} q_\phi(\boldsymbol{S}, \boldsymbol{c}|\boldsymbol{X}) \log p_\theta(\boldsymbol{X}|\boldsymbol{S}, \boldsymbol{c}) d\boldsymbol{S} d\boldsymbol{c} \\
=& \int_{\boldsymbol{c}} q_{\phi_{\boldsymbol{c}}}(\boldsymbol{c}|\boldsymbol{X}) \int_{\boldsymbol{S}} q_{\phi_{\boldsymbol{S}}}(\boldsymbol{S}|\boldsymbol{X}) \log p_\theta(\boldsymbol{X}|\boldsymbol{S}, \boldsymbol{c}) d\boldsymbol{S} d\boldsymbol{c} \\
=& \int_{\boldsymbol{c}} q_{\phi_{\boldsymbol{c}}}(\boldsymbol{c}|\boldsymbol{X}) \sum_{j=1}^{M} \int_{\boldsymbol{s}_j} q_{\phi_{\boldsymbol{s}_j}}(\boldsymbol{s}_j|\boldsymbol{x}_j) \log p_\theta(\boldsymbol{x}_j|\boldsymbol{s}_j, \boldsymbol{c}) d\boldsymbol{s}_j d\boldsymbol{c} \\
=& \sum_{j=1}^{M} \int_{\boldsymbol{c}} q_{\phi_{\boldsymbol{c}}}(\boldsymbol{c}|\boldsymbol{X}) \int_{\boldsymbol{s}_j} q_{\phi_{\boldsymbol{s}_j}}(\boldsymbol{s}_j|\boldsymbol{x}_j) \log p_\theta(\boldsymbol{x}_j|\boldsymbol{s}_j, \boldsymbol{c}) d\boldsymbol{s}_j d\boldsymbol{c} \\
=& \sum_{j=1}^{M} E_{q_{\phi_{\boldsymbol{c}}}(\boldsymbol{c}|\boldsymbol{X})}[E_{q_{\phi_{\boldsymbol{s}_j}}(\boldsymbol{s}_j|\boldsymbol{x}_j)}[\log p_\theta(\boldsymbol{x}_j|\boldsymbol{s}_j, \boldsymbol{c})]]
\end{aligned}
\tag{17}
$$

From Equation (17), the final form of $\widetilde{\mathcal{L}}(\theta, \phi; \boldsymbol{X})$ follows directly:

$$\widetilde{\mathcal{L}}(\theta, \phi; \boldsymbol{X}) = \sum_{j=1}^{M} E_{q_{\phi_c}(\boldsymbol{c}|\boldsymbol{X})}[E_{q_{\phi_{s_j}}(\boldsymbol{s}_j|\boldsymbol{x}_j)}[\log p_\theta(\boldsymbol{x}_j|\boldsymbol{s}_j, \boldsymbol{c})]]$$

$$- JS_{\boldsymbol{\pi}}^{M+1}(\{q_{\phi_{c_j}}(\boldsymbol{c}|\boldsymbol{x}_j)\}_{j=1}^{M}, p_\theta(\boldsymbol{c})) - \sum_{j=1}^{M} KL(q_{\phi_{s_j}}(\boldsymbol{s}_j|\boldsymbol{x}_j)||p_\theta(\boldsymbol{s}_j)) \qquad (18)$$

### B.5 JS-divergence as intermodality divergence

Utilizing the JS-divergence as regularization term as proposed in this work has multiple effects on the training procedure. The first is the introduction of the dynamic prior as described in the main paper. A second effect is the minimization of the intermodality-divergence. The intermodality-divergence is the difference of the posterior approximations between modalities. For a coherent generation, the posterior approximations of all modalities should be similar such that - if only a single modality is given - the decoders of the missing data types are able to generate coherent samples. Using the JS-divergence as regularization term keeps the unimodal posterior approximations similar to its mixture distribution. This can be compared to minimizing the divergence between the unimodal distributions and its mixture which again can be seen as an efficient approximation of minimizing the $M^2$ pairwise unimodal divergences, the intermodality-divergences. Wu and Goodman [9] report problems in optimizing the unimodal posterior approximations. These problems lead to diverging posterior approximations which again results in bad coherence for missing data generation. Diverging posterior approximations cannot be handled by the decoders of the missing modality.

## C Experiments

In this section we describe the architecture and implementation details of the different experiments. Additionally, we show more results and ablation studies.

### C.1 Modality-specific Sub-Spaces

Similar to Equation (16), we derive the factorized ELBO for the MVAE Wu and Goodman [9] and MMVAE [7] model. It follows:

$$\mathcal{L}_{\text{PoE}}(\theta, \phi; \boldsymbol{X}) = \sum_{j=1}^{M} E_{q_{\phi_c}(\boldsymbol{c}|\boldsymbol{X})}[E_{q_{\phi_{s_j}}(\boldsymbol{s}_j|\boldsymbol{x}_j)}[\log p_\theta(\boldsymbol{x}_j|\boldsymbol{s}_j, \boldsymbol{c})]]$$

$$- \text{PoE}(\{q_{\phi_{c_j}}(\boldsymbol{c}|\boldsymbol{x}_j)\}_{j=1}^{M}, p_\theta(\boldsymbol{c})) - \sum_{j=1}^{M} KL(q_{\phi_{s_j}}(\boldsymbol{s}_j|\boldsymbol{x}_j)||p_\theta(\boldsymbol{s}_j)) \qquad (19)$$

$$\mathcal{L}_{\text{MoE}}(\theta, \phi; \boldsymbol{X}) = \sum_{j=1}^{M} E_{q_{\phi_c}(\boldsymbol{c}|\boldsymbol{X})}[E_{q_{\phi_{s_j}}(\boldsymbol{s}_j|\boldsymbol{x}_j)}[\log p_\theta(\boldsymbol{x}_j|\boldsymbol{s}_j, \boldsymbol{c})]]$$

$$- \text{MoE}(\{q_{\phi_{c_j}}(\boldsymbol{c}|\boldsymbol{x}_j)\}_{j=1}^{M}, p_\theta(\boldsymbol{c})) - \sum_{j=1}^{M} KL(q_{\phi_{s_j}}(\boldsymbol{s}_j|\boldsymbol{x}_j)||p_\theta(\boldsymbol{s}_j)) \qquad (20)$$

The above ELBOs $\mathcal{L}_{\text{PoE}}(\theta, \phi; \boldsymbol{X})$ and $\mathcal{L}_{\text{MoE}}(\theta, \phi; \boldsymbol{X})$ are used in the experiments section for all models with a (MS)-suffix.

### C.2 Evaluation

First we describe the architectures and models used for evaluating classification accuracies.

Table 1: Layers for MNIST and SVHN classifiers. For MNIST and SVHN, every convolutional layer is followed by a ReLU activation function. For SVHN, every convolutional layer is followed by a dropout layer (dropout probability = 0.5). Then, batchnorm is applied followed by a ReLU activation function. The output activation is a sigmoid function for both classifiers. Specifications (Spec.) name kernel size, stride, padding and dilation.

| | MNIST | | | | | SVHN | | | |
|---|---|---|---|---|---|---|---|---|---|
| Layer | Type | #F. In | #F. Out | Spec. | Layer | Type | #F. In | #F. Out | Spec. |
| 1 | $conv_{2d}$ | 1 | 32 | (4, 2, 1, 1) | 1 | $conv_{2d}$ | 1 | 32 | (4, 2, 1, 1) |
| 2 | $conv_{2d}$ | 32 | 64 | (4, 2, 1, 1) | 2 | $conv_{2d}$ | 32 | 64 | (4, 2, 1, 1) |
| 3 | $conv_{2d}$ | 64 | 128 | (4, 2, 1, 1) | 3 | $conv_{2d}$ | 64 | 64 | (4, 2, 1, 1) |
| 4 | linear | 128 | 10 | | 4 | $conv_{2d}$ | 64 | 128 | (4, 2, 0, 1) |
| | | | | | 5 | linear | 128 | 10 | |

Table 2: Layers for the Text classifier for MNIST-SVHN-Text. The text classifier consists of residual layers as described by He et al. [3] for 1d-convolutions. The output activation is a sigmoid function. Specifications (Spec.) name kernel size, stride, padding and dilation.

| Layer | Type | #F. In | #F. Out | Spec. |
|---|---|---|---|---|
| 1 | $conv_{1d}$ | 71 | 128 | (1, 1, 1, 1) |
| 2 | $residual_{1d}$ | 128 | 192 | (4, 2, 1, 1) |
| 3 | $residual_{1d}$ | 192 | 256 | (4, 2, 1, 1) |
| 4 | $residual_{1d}$ | 256 | 256 | (4, 2, 1, 1) |
| 5 | $residual_{1d}$ | 256 | 128 | (4, 2, 0, 1) |
| 6 | linear | 128 | 10 | |

### C.2.1 Latent Representations

To evaluate the learned latent representations, we use a simple logistic regression classifier without any regularization. We use a predefined model by scikit-learn [6]. Every linear classifier is trained on a single batch of latent representations. For simplicity, we always take the last batch of the training set to train the classifier. The trained linear classifier is then used to evaluate the latent representations of all samples in the test set.

### C.2.2 Generated Samples

To evaluate generated samples regarding their content coherence, we classify them according to the attributes of the dataset. In case of missing data, the estimated data types must coincide with the available ones according to the attributes present in the available data types. In case of random generation, generated samples of all modalities must be coincide with each other. To evaluate the coherence of generated samples, classifiers are trained for every modality. If the detected attributes for all involved modalities are the same, the generated samples are called coherent. For all modalities, classifiers are trained on the original, unimodal training set. The architectures of all used classifiers can be seen in Tables 1 to 3.

## C.3 MNIST-SVHN-Text

### C.3.1 Text Modality

To have an additional modality, we generate text from labels. As a single word is quite easy to learn, we create strings of length 8 where everything is a blank space except the digit-word. The starting position of the word is chosen randomly to increase the difficulty of the learning task. Some example strings can be seen in Table 4.

Table 3: CelebA Classifiers. The image classifier consists of residual layers as described by He et al. [3] followed by a linear layer which maps to 40 output neurons representing the 40 attributes. The text classifier also uses residual layers, but for 1d-convolutions. The output activation is a sigmoid function for both classifiers. Specifications (Spec.) name kernel size, stride, padding and dilation.

| | | Image | | | | | Text | | | |
| Layer | Type | #F. In | #F. Out | Spec. | Layer | Type | #F. In | #F. Out | Spec. |
|---|---|---|---|---|---|---|---|---|---|
| 1 | $\text{conv}_{2d}$ | 3 | 128 | (3, 2, 1, 1) | 1 | $\text{conv}_{1d}$ | 71 | 128 | (3, 2, 1, 1) |
| 2 | $\text{res}_{2d}$ | 128 | 256 | (4, 2, 1, 1) | 2 | $\text{res}_{1d}$ | 128 | 256 | (4, 2, 1, 1) |
| 3 | $\text{res}_{2d}$ | 256 | 384 | (4, 2, 1, 1) | 3 | $\text{res}_{1d}$ | 256 | 384 | (4, 2, 1, 1) |
| 4 | $\text{res}_{2d}$ | 384 | 512 | (4, 2, 1, 1) | 4 | $\text{res}_{1d}$ | 384 | 512 | (4, 2, 1, 1) |
| 5 | $\text{res}_{2d}$ | 512 | 640 | (4, 2, 0, 1) | 5 | $\text{res}_{1d}$ | 512 | 640 | (4, 2, 1, 1) |
| 6 | linear | 640 | 40 | | 6 | $\text{residual}_{1d}$ | 640 | 768 | (4, 2, 1, 1) |
| | | | | | 7 | $\text{residual}_{1d}$ | 768 | 896 | (4, 2, 0, 1) |
| | | | | | 8 | linear | 896 | 40 | |

Table 4: Example strings to create an additional text modality for the MNIST-SVHN-Text dataset. This results in triples of texts and two different image modalities.

```
                                         six
                                     eight
                     three
                         five
                                  nine
                     zero
                                          four
                                     three
                 seven
                     five
```

Table 5: MIST: Encoder and Decoder Layers. Every layer is followed by ReLU activation function. Layers 3a and 3b of the encoder are needed to map to $\mu$ and $\sigma^2 I$ of the approximate posterior distribution.

| | | Encoder | | | | Decoder | |
| Layer | Type | # Features In | # Features Out | Layer | Type | # Features In | # Features Out |
|---|---|---|---|---|---|---|---|
| 1 | linear | 784 | 400 | 1 | linear | 20 | 400 |
| 2a | linear | 400 | 20 | 2 | linear | 400 | 784 |
| 2b | linear | 400 | 20 | | | | |

Table 6: SVHN: Encoder and Decoder Layers. The specifications name kernel size, stride, padding and dilation. All layers are followed by a ReLU activation function.

| | | Encoder | | | | | Decoder | | |
| Layer | Type | #F. In | #F. Out | Spec. | Layer | Type | #F. In | #F. Out | Spec. |
|---|---|---|---|---|---|---|---|---|---|
| 1 | $\text{conv}_{2d}$ | 3 | 32 | (4, 2, 1, 1) | 1 | linear | 20 | 128 | |
| 2 | $\text{conv}_{2d}$ | 32 | 64 | (4, 2, 1, 1) | 2 | $\text{conv}_{2d}^{T}$ | 128 | 64 | (4, 2, 0, 1) |
| 3 | $\text{conv}_{2d}$ | 64 | 64 | (4, 2, 1, 1) | 3 | $\text{conv}_{2d}^{T}$ | 64 | 64 | (4, 2, 1, 1) |
| 4 | $\text{conv}_{2d}$ | 64 | 128 | (4, 2, 0, 1) | 4 | $\text{conv}_{2d}^{T}$ | 64 | 32 | (4, 2, 1, 1) |
| 5a | linear | 128 | 20 | | 5 | $\text{conv}_{2d}^{T}$ | 32 | 3 | (4, 2, 1, 1) |
| 5b | linear | 128 | 20 | | | | | | |

Table 7: Text for MNIST-SVHN-Text: Encoder and Decoder Layers. The specifications name kernel size, stride, padding and dilation. All layers are followed by a ReLU activation function.

| | | Encoder | | | | | Decoder | | | |
|---|---|---|---|---|---|---|---|---|---|---|
| Layer | Type | #F. In | #F. Out | Spec. | Layer | Type | #F. In | #F. Out | Spec. |
| 1 | $\text{conv}_{1d}$ | 71 | 128 | (1, 1, 0, 1) | 1 | linear | 20 | 128 | |
| 2 | $\text{conv}_{1d}$ | 128 | 128 | (4, 2, 1, 1) | 2 | $\text{conv}_{1d}^T$ | 128 | 128 | (4, 1, 0, 1) |
| 3 | $\text{conv}_{1d}$ | 128 | 128 | (4, 2, 0, 1) | 3 | $\text{conv}_{1d}^T$ | 128 | 128 | (4, 2, 1, 1) |
| 4a | linear | 128 | 20 | | 4 | $\text{conv}_{1d}^T$ | 128 | 71 | (1, 1, 0, 1) |
| 4b | linear | 128 | 20 | | | | | | |

### C.3.2   Implementation Details

For MNIST and SVHN, we use the network architectures also utilized by [7] (see Table 5 and Table 6). The network architecture used for the Text modality is described in Table 7. For all encoders, the last layers named a and b are needed to map to $\mu$ and $\sigma^2 \boldsymbol{I}$ of the posterior distribution. In case of modality-specific sub-spaces, there are four last layers to map to $\mu_s$ and $\sigma_s^2 \boldsymbol{I}$ and $\mu_c$ and $\sigma_c^2 \boldsymbol{I}$.

To enable a joint latent space, all modalities are mapped to have a 20 dimensional latent space (like in Shi et al. [7]). For a latent space with modality-specific and -independent sub-spaces, this restriction is not needed anymore. Only the modality-invariant sub-spaces of all data types must have the same number of latent dimensions. Nevertheless, we create modality-specific sub-spaces of the same size for all modalities. For the results reported in the main text, we set it to 4. To have an equal number of parameters as in the experiment with only a shared latent space, we set the shared latent space to 16 dimensions. This allows for a fair comparison between the two variants regarding the capacity of the latent space. See appendix C.3.5 and Figure 3 for a detailed comparison regarding the size of the modality specific-subspaces. Modality-specific sub-spaces are a possibility to account for the difficulty of every data type.

The image modalities are modelled with a Laplace likelihood and the text modality is modelled with a categorical likelihood. The likelihood-scaling is done according to the data size of every modality. The weight of the largest data type, i.e. SVHN, is set to 1.0. The weight for MNIST is given by $size(SVHN)/size(MNIST)$ and the text weight by $size(SVHN)/size(Text)$. This scaling scheme stays the same for all experiments. The weight of the unimodal posteriors are equally weighted to form the joint distribution. This is true for MMVAE and mmJSD. For MVAE, the posteriors are weighted according to the inverse of their variance. For mmJSD, all modalities and the pre-defined distribution are weighted 0.25. We keep this for all experiments reported in the main paper. See appendix C.3.6 and Figure 4 for a more detailed analysis of distribution weights.

For all experiments, we set $\beta$ to 5.0. For all experiments with modality-specific subspaces, the $\beta$ for the modality-specific subspaces is set equal to the number of modalities, i.e. 3. Additionally, the $\beta$ for the text modality is set to 5.0, for the other 2 modalities it is set to 1.0. The evaluation of different $\beta$-values shows the stability of the model according to this hyper-parameter (see Figure 1).

All unimodal posterior approximations are assumed to be Gaussian distributed $\mathcal{N}(\boldsymbol{\mu}_\nu(\boldsymbol{x}_\nu), \boldsymbol{\sigma}_\nu^2(\boldsymbol{x}_\nu)\boldsymbol{I})$, as well as the pre-defined distribution $p_\theta(\boldsymbol{z})$ which is defined as $\mathcal{N}(\boldsymbol{0}, \boldsymbol{I})$.

For training, we use a batch size of 256 and a starting learning rate of 0.001 together with an ADAM optimizer [5]. We pair every MNIST image with 20 SVHN images which increases the dataset size by a factor of 20. We train our models for 50 epochs in case of a shared latent space only. In case of modality-specific subspaces we train the models for 100 epochs. This is the same for all methods.

### C.3.3   Qualitative Results

Figure 2 shows qualitative results for the random generation of MNIST and SVHN samples.

### C.3.4   Comparison to Shi et al.

The results reported in Shi et al. [7]'s paper with the MMVAE model rely heavily on importance sampling (IS) (as can be seen by comparing to the numbers of a model without IS reported in their

(a) Latent Representation Classification

(b) Generation Coherence

(c) Quality of Samples

Figure 1: Comparison of different $\beta$ values with respect to generation coherence, quality of latent representations (measured in accuracy) and quality of generated samples (measured in precision-recall for generative models).

(a) MVAE: MNIST

(b) MMVAE: MNIST

(c) mmJSD: MNIST

(d) MVAE: SVHN

(e) MMVAE: SVHN

(f) mmJSD: SVHN

Figure 2: Qualitative results for random generation.

Table 8: Comparison of training times on the MNIST-SVHN-Text dataset. (I=30) names the model with 30 importance samples.

| MODEL | #EPOCHS | RUNTIME |
|---|---|---|
| MVAE | 50 | 3H 01MIN |
| MMVAE | 50 | 2H 01MIN |
| MMVAE (I=30) | 30 | 15H 15MIN |
| MMJSD | 50 | 2H 16MIN |
| MVAE (MS) | 100 | 6H 15MIN |
| MMVAE (MS) | 100 | 4H 10MIN |
| MMJSD (MS) | 100 | 4H 36MIN |

Table 9: Classification accuracy of the learned latent representations using a linear classifier. We evaluate all subsets of modalities for which we use the following abbreviations: M: MNIST; S: SVHN; T: Text; M,S: MNIST and SVHN; M,T: MNIST and Text; S,T: SVHN and Text; Joint: all modalities. (MS) names the models with modality-specific latent subspaces. (I=30) names the model with 30 importance samples.

| MODEL | M | S | T | M,S | M,T | S,T | JOINT |
|---|---|---|---|---|---|---|---|
| MMVAE | 0.96 | 0.81 | **0.99** | 0.89 | 0.97 | 0.90 | 0.93 |
| MMVAE (I=30) | 0.92 | 0.67 | **0.99** | 0.80 | 0.96 | 0.83 | 0.86 |
| **MMJSD** | 0.97 | 0.82 | **0.99** | 0.93 | **0.99** | 0.92 | 0.98 |
| MMVAE (MS) | 0.96 | 0.81 | **0.99** | 0.89 | 0.98 | 0.91 | 0.92 |
| **MMJSD (MS)** | **0.98** | **0.85** | **0.99** | **0.94** | 0.98 | **0.94** | **0.99** |

appendix). The IS-based objective [2] is a different objective and difficult to compare to models without an IS-based objective. Hence, to have a fair comparison between all models we compared all models without IS-based objective in the main paper. The focus of the paper was on the different joint posterior approximation functions and the corresponding ELBO which should reflect the problems of a multimodal model.

For completeness we compare the proposed model to the IS-based MMVAE model here in the appendix. Table 8 shows the training times for the different models. Although the MMVAE (I=30) only needs 30 training epochs for convergence, these 30 epochs take approximately 3 times as long as for the other models without importance sampling. (I=30) names the model with 30 importance samples. What is also adding up to the training time for the MMVAE (I=30) model is the $M^2$ paths through the decoder. The MMVAE model and mmJSD need approximately the same time until training is finished. MVAE takes longer as the training objective is a combination of ELBOs instead of a single objective.

Table 10: Classification accuracy of generated samples on MNIST-SVHN-Text. In case of conditional generation, the letter above the horizontal line indicates the modality which is generated based on the different sets of modalities below the horizontal line. (I=30) names the model with 30 importance samples.

| MODEL | RANDOM | M | | | S | | | T | | |
|---|---|---|---|---|---|---|---|---|---|---|
| | | S | T | S,T | M | T | M,T | M | S | M,S |
| MMVAE (I=30) | 0.60 | 0.71 | **0.99** | 0.85 | 0.76 | 0.68 | 0.72 | 0.95 | 0.73 | 0.84 |
| MMVAE | 0.54 | **0.82** | **0.99** | 0.91 | 0.32 | 0.30 | 0.31 | 0.96 | **0.83** | 0.90 |
| **MMJSD** | 0.60 | **0.82** | **0.99** | **0.95** | 0.37 | 0.36 | 0.48 | **0.97** | **0.83** | **0.92** |
| MMVAE (MS) | **0.67** | 0.77 | 0.97 | 0.86 | 0.88 | **0.93** | 0.90 | 0.82 | 0.70 | 0.76 |
| **MMJSD (MS)** | 0.66 | 0.80 | 0.97 | 0.93 | **0.89** | **0.93** | **0.92** | 0.92 | 0.79 | 0.86 |

Table 11: Test set log-likelihood on MNIST-SVHN-Text. We report the log-likelihood of the joint generative model $p_\theta(\boldsymbol{X})$. (I=30) names the model with 30 importance samples.

| MODEL | $\boldsymbol{X}$ |
|---|---|
| MVAE | **-1864** |
| MMVAE (I=30) | -1891 |
| MMVAE | -1916 |
| MMJSD | -1961 |
| MVAE (MS) | -1870 |
| MMVAE (MS) | -1893 |
| MMJSD (MS) | -1900 |

Tables 9, 10 and 11 show that the models without any importance samples achieve state-of-the-art performance compared to the MMVAE model using importance samples. Using modality-specific subspaces seems to have a similar effect towards test set log-likelihood performance as using importance samples with a much lower impact on computational efficiency as it can be seen in the comparison of training times in Table 8.

### C.3.5 Modality-Specific Subspaces

The introduction of modality-specific subspaces introduces an additional degree of freedom. In Figure 3, we show a comparison of different modality-specific subspace sizes. The size is the same for all modalities. Also, the total number of latent dimensions is constant, i.e. the number of dimensions in the modality-specific subspaces is subtracted from the shared latent space. If we have modality-specific latent spaces of size 2, the shared latent space is of size 18. This allows to ensure that the capacity of latent spaces stays constant. Figure 3 shows that the introduction of modality-specific subspaces only has minor effect on the quality of learned representations, despite the lower number of dimensions in the shared space. Generation coherence suffers with increasing number of modality-specific dimensions, but the quality of samples improves. We guess that the coherence becomes lower due to information which is shared between modalities but encoded in modality-specific spaces. In future work, we are interested in finding better schemes to identify shared and modality-specific information.

### C.3.6 Weight of predefined distribution in JS-divergence

We empirically analyzed the influence of different weights of the pre-defined distribution $p_\theta(\boldsymbol{z})$ in the JS-divergence. Figure 4 shows the results. We see the constant performance regarding the latent representations and the quality of samples. In future work we would like to study the drop in performance regarding the coherence of samples if the weight of the pre-defined distribution $p_\theta(\boldsymbol{z})$ is around 0.4.

## C.4 CelebA

### C.4.1 Bimodal Dataset

Every face in the dataset is labelled with 40 attributes. For the text modality, we create text strings from these attributes. The text modality is a concatenation of available attributes into a comma-separated list. Underline characters are replaced by a blank space. We create strings of length 256 (which is the maximum string length possible following described rules). If a given face has only a small number of attributes which would result in a short string, we fill the remaining space with the asterix character $*$. Table 12 shows examples of strings.

### C.4.2 Implementation Details

For the CelebA experiments, we switched to a ResNet architecture [3] for encoders and decoders of image and text modality due to the difficulty of the dataset. The specifications of the individual layers for the image and text networks can be found in Tables 13 and 14. The image modality is modelled with a Laplace likelihood and a Gaussian distributed posterior approximation. The text modality is

(a) Latent Representation Classification

(b) Generation Coherence

(c) Quality of Samples

Figure 3: Comparison of different modality-specific latent space sizes for the proposed mmJSD objective.

Table 12: Examples of strings we created to have a bimodal version of CelebA which results in pairs of images and texts. For illustrative reasons we dropped the asterix characters.

bags under eyes, chubby, eyeglasses, gray hair, male, mouth slightly open, oval face, sideburns, smiling, straight hair
big nose, male, no beard, young
attractive, big nose, black hair, bushy eyebrows, high cheekbones, male, mouth slightly open, no beard, oval face, smiling, young
5 o clock shadow, bags under eyes, big nose, bushy eyebrows, chubby, double chin, gray hair, high cheekbones, male, mouth slightly open, no beard, smiling, straight hair, wearing necktie
arched eyebrows, attractive, bangs, black hair, heavy makeup, high cheekbones, mouth slightly open, no beard, pale skin, smiling, straight hair, wearing lipstick, young
attractive, brown hair, bushy eyebrows, high cheekbones, male, no beard, oval face, smiling, young
attractive, high cheekbones, no beard, oval face, smiling, wearing lipstick, young
attractive, blond hair, heavy makeup, high cheekbones, mouth slightly open, no beard, oval face, smiling, wearing lipstick, young
attractive, brown hair, heavy makeup, no beard, oval face, pointy nose, straight hair, wearing lipstick, young
5 o clock shadow, bags under eyes, big nose, brown hair, male, mouth slightly open, smiling, young
attractive, brown hair, heavy makeup, high cheekbones, mouth slightly open, no beard, oval face, pointy nose, smiling, wavy hair, wearing earrings, wearing lipstick, young
attractive, bangs, blond hair, heavy makeup, high cheekbones, mouth slightly open, no beard, oval face, smiling, wavy hair, wearing earrings, wearing lipstick, young

modelled with a categorical likelihood and a Gaussian distributed posterior approximation. Their likelihoods are weighted according to the data size with the image likelihood being set to 1.0. The text likelihood is scaled according to $size(Img)/size(Text)$. The global $\beta$ is set to 2.5 and the $\beta_S$ of the modality-specific subspaces again to the number of modalities, i.e. 2. The shared as well as the modality-specific latent spaces consist all of 32 dimensions. For training, we used a batch size of 256. We use ADAM as optimizer [5] with a starting learning rate of 0.001. We trained our model for 100 epochs.

### C.4.3 Results

In Figure 5, we show randomly generated images sampled from the joint latent distribution. Table 15 shows the corresponding text samples of the first row in Figure 5. Figures 6 and 7 show quantitative results which demonstrate the difficulty of this dataset. Figure 6 show classification accuracies of the latent representation for the different attributes. Because of the imbalanced nature of some attributes, we report the average precision. This figure demonstrates the difficulty to learn a good latent representation for all attributes. A similar pattern can be seen in Figure 7 which shows the classification performance of generated samples according to the different attributes. The distribution

(a) Latent Representation Classification

(b) Generation Coherence

(c) Quality of Samples

Figure 4: Comparison of different weights for the pre-defined distribution $p_\theta(z)$ in the JS-divergence.

Table 13: CelebA Image: Encoder and Decoder Layers. The specifications name kernel size, stride, padding and dilation. res names a residual block.

| | Encoder | | | | | Decoder | | | |
|---|---|---|---|---|---|---|---|---|---|
| Layer | Type | #F. In | #F. Out | Spec. | Layer | Type | #F. In | #F. Out | Spec. |
| 1 | $\text{conv}_{2d}$ | 3 | 128 | (3, 2, 1, 1) | 1 | linear | 64 | 640 | |
| 2 | $\text{res}_{2d}$ | 128 | 256 | (4, 2, 1, 1) | 2 | $\text{res}_{2d}^{T}$ | 640 | 512 | (4, 1, 0, 1) |
| 3 | $\text{res}_{2d}$ | 256 | 384 | (4, 2, 1, 1) | 3 | $\text{res}_{2d}^{T}$ | 512 | 384 | (4, 1, 1, 1) |
| 4 | $\text{res}_{2d}$ | 384 | 512 | (4, 2, 1, 1) | 4 | $\text{res}_{2d}^{T}$ | 384 | 256 | (4, 1, 1, 1) |
| 5 | $\text{res}_{2d}$ | 512 | 640 | (4, 2, 1, 1) | 5 | $\text{res}_{2d}^{T}$ | 256 | 128 | (4, 1, 1, 1) |
| 6a | linear | 640 | 32 | | 6 | $\text{conv}_{2d}^{T}$ | 128 | 3 | (3, 2, 1, 1) |
| 6b | linear | 640 | 32 | | | | | | |

Table 14: CelebA Text: Encoder and Decoder Layers. The specifications name kernel size, stride, padding and dilation. res names a residual block.

| | Encoder | | | | | Decoder | | | |
|---|---|---|---|---|---|---|---|---|---|
| Layer | Type | #F. In | #F. Out | Spec. | Layer | Type | #F. In | #F. Out | Spec. |
| 1 | $\text{conv}_{1d}$ | 71 | 128 | (3, 2, 1, 1) | 1 | linear | 64 | 896 | |
| 2 | $\text{res}_{1d}$ | 128 | 256 | (4, 2, 1, 1) | 2 | $\text{res}_{1d}^{T}$ | 640 | 640 | (4, 2, 0, 1) |
| 3 | $\text{res}_{1d}$ | 256 | 384 | (4, 2, 1, 1) | 3 | $\text{res}_{1d}^{T}$ | 640 | 640 | (4, 2, 1, 1) |
| 4 | $\text{res}_{1d}$ | 384 | 512 | (4, 2, 1, 1) | 4 | $\text{res}_{1d}^{T}$ | 640 | 512 | (4, 2, 1, 1) |
| 5 | $\text{res}_{1d}$ | 512 | 640 | (4, 2, 1, 1) | 5 | $\text{res}_{1d}^{T}$ | 512 | 384 | (4, 2, 1, 1) |
| 6 | $\text{res}_{1d}$ | 640 | 640 | (4, 2, 1, 1) | 6 | $\text{res}_{1d}^{T}$ | 384 | 256 | (4, 2, 1, 1) |
| 7 | $\text{res}_{1d}$ | 640 | 640 | (4, 2, 0, 1) | 7 | $\text{res}_{1d}^{T}$ | 256 | 128 | (4, 2, 1, 1) |
| 8a | linear | 640 | 32 | | 8 | $\text{conv}_{1d}^{T}$ | 128 | 71 | (3, 2, 1, 1) |
| 8b | linear | 640 | 32 | | | | | | |

Figure 5: Randomly generated CelebA images sampled from the joint latent space of the proposed model.

Table 15: Randomly generated CelebA strings sampled from the joint latent space of the proposed model. The strings correspond to the first row of images in Figure 5. We cut after the remaining asterix characters for illustrative reasons.

5 o clock shadow, arched eyebrows, attig lips, blldn ha, big n ee, basd, your, ho beari, ntraight hair, wyanose, smiling, wearins eactt* *** blg******ing****
bangs, big lips, brown hair, gray hair, male, no beard, woung**********************************************************************************
arched eyebrows, attractive, bengy blbcows, heuvy eabones, mouth slig tlyr, narrow eyes, no beard, smiling, posniyhngiewavy hang, young*******
bangs, big lips, black hair, high cheekbones, mouth slightly open, no beard, pale skin, wavy hang, young**********************************
big lips, big nose, black hair, bushy ey, high cheekbones, narrow eyes, noface, pointy nose, smiling, wavy hair, young************************
bags under eyes, mouth slightly open, no beard, smiface, straight hairsmilirair,traight hair, young****************************************
attractive, blond hair, brown hhigh chee aoses, mouth slightly open, no beard, oval facg, young***************************************
arched eyebrows, bags under eyes, blackose, black h ir, ch ebes, narrow eyep, no eard, wavy hair, wearing lipstick, young*********************
big nose, blond eyebrows, no bmale, s, no beard, wavy hair, young***************************************************************
attractive, black hair, heavy makeup, high cheekbones, no beard, smiling, wearing lipstick, young***********************************
5 o clock shadow, bags under eyes, bald, mase, mou hegh arrow eyes, no beard, straight hair, wearing lipstick, young*************************
black hair, blurry, brown hair,p, o albeard, smiling**************************************************************************
attractive, black hair, brown hair, maatbe, mals, no beard, rosy ling, w smiling****************************************************
arched eyebrows, attractive, brown hair, bl ngwe, weari, youtg***************************************************************
big lips, eyeglasses, high, no bea d, yeang, young*******************************************************************
bangs, brown hair, byehlasses, ws,vmouth sl, no beard, oval facd, smiling, wearing lipstick, young**********************************

over classification performances of latent representations and conditionally generated samples is similar. This pattern gives further evidence on the importance of a good latent representation for coherent generation in case of missing data. Additionally, Figure 6 and 7 show the superior performance of the proposed mmJSD objective with respect to almost all attributes.

(a) Img

(b) Text

(c) Joint

Figure 6: Classification of learned representations on CelebA. We report the average precision (higher is better). The difficulty of learning the individual attributes can be seen by the difference in classification performance across attributes. On the other hand, performance distribution over attributes is similar for both modalities. For all subsets of modalities, the proposed mmJSD objective outperforms previous work.

(a) Img

(b) Text

Figure 7: Classification accuracies of generated samples on CelebA. Coherent generation is mostly only possible if a linearly separable representation of an attribute is learned (see Figure 6). The proposed mmJSD method achieves state-of-the-art or superior performance in the generation of both modalities. Img stands for images which are generated conditioned on text sample, Text for texts which are generated based on image samples.