[Reviews · NeurIPS 2020]

Review 1

Summary and Contributions: This paper proposed to use JS divergence as a new regularizer for inference process of multimodal generation. The applied JS divergence is able to model both unimodal and joint multimodal posteriors which is derived from the introduced dynamic prior. The authors also proved that the new objective optimizes an ELBO. In experiments, the proposed method was compared with two prior work MVAE and MMVAE on discrimination, generation task, and NLL value.

Strengths: Being scalable, handling missing modals are important for multimodal generation. This paper focused on these points and proposed the JS divergence as a new regularizer for multiple modal’s posterior. Instead of using a simple prior which regularizes every unimodal approximation, the introduced dynamic prior is expected to also capture the joint information once it is well defined, such as MoE or PoE. As the inequality of JS distance and KL distance holds, the modified multimodal objective ELBO is a lower bound to the original one (Eq. 2 in the paper). Experiments show that MMJSD with PoE prior is usually better than MVAE and MMVAE, while its NLL value is always larger.

Weaknesses: 1. The first weakness of this work is that the wish-list presented in the Introduction is a bit wider than the real techniques proposed by this work, because the key difference of this work lies in the dynamic prior. The three properties were mentioned and basically solved by previous work like reference [21] and [27]. Especially for Missing data, this could be seen as the main application of multimodal generation, i.e., cross-modal generation. I feel it is fine to present in the Introduction like this, but if you want to stress them, they should be closely related to the contributions. 2. I understand that dynamic prior defined with MoE type is clearly different from MMVAE. However, to have a closed form solution, MoE is introduced as the third-party distribution which constructs the JS divergence and all the distribution here should be exponential, like Gaussian used in this paper. The express might be the limitation in this case, so is it possible to directly constrain z alternatively? 3. From my understanding, the proposed JS divergence captures the joint information (suppose the proposed dynamic prior could achieve this) by sacrificing some approximation to the original ELBO. So visualized latent representation might be needed to further demonstration. 4. Comparing Eqs. 6 and 10, we can see that there are M+1 priors need to be defined in Eq. 10 while only one in Eq. 6. Will it be a practical issue in real implementation? 4. As MoE is said to perform better in reference [21], like not suffering from the overconfident experts or training better when all modals are available, I felt confused why JSD with PoE is used throughout the experiments without clear motivations. 5. Note that importance sampling is used in MMVAE which helps develop a tighter bound. Therefore, its NLL is supposed to be larger theoretically. The authors claimed they do not have closed form solution for KL-divergence which causes a less efficient computation. Eq. 6 seems to be an indirect evidence, but it is better to provide experimental results to support this claim.

Correctness: I did not find any mistakes, but in my view the proof of Lemma 2 is trivial.

Clarity: This paper is well organized and the problem is clearly stated. The main techniques are also well presented and readable.

Relation to Prior Work: Reference [21] and [27] are mostly related to this work. Some differences need to be further clarified as mentioned in weaknesses.

Reproducibility: Yes

Additional Feedback: After reading the rebuttal, the authors has addressed the concerns I commented before, hence I tend to maintain the previous rating of "Marginally above the acceptance threshold".


Review 2

Summary and Contributions: This paper proposes a new training objective for a generative model of multiple data modalities. In particular the objective is a variational lower bound to the log likelihood of a VAE model, utilizing the Jensen-Shannon divergence between the various latent distributions rather than the usual KL divergence between the approximate posterior and the prior. The objective is motivated and derived, and then demonstrated on a range of multi-modal generative tasks.

Strengths: The paper lays out solid motivation to establish a new training objective for multi-modal unsupervised learning. That is, computational efficiency and the ability to learn effectively from data with a subset of modalities available. These criteria appear to be satisfied by the proposed training objective, which to my knowledge is novel. The proposed lower bound for the log likelihood is itself relatively straightforward, and bears resemblance to the usual ELBO. This is advantageous over other multi-modal schemes which can rely on somewhat unprincipled techniques such as parameter sharing across certain neural network modules. The experimental results shown largely seem to demonstrate the effectiveness of the method, comparing favourably to the closest architecture - the MMVAE. The results are shown over a good variety of tasks and the image-text modalities, which are sufficiently different to indicate that the method is general.

Weaknesses: In this work the latent space is decomposed into modality-specific and shared latent spaces. However, it is not shown experimentally how the functions of these two latents differ. It would have been interesting to see how traversing the global latent affected different modalities, and vice versa for the modality-specific latents. Without some kind of verification, it is not obvious that the given factorization yields any benefit. It is also unclear exactly why the authors choose to the use the Product-of-Experts approach rather than the Mixture-of-Experts. Indeed, it has been noted in the literature before (e.g. in both the MVAE and MMVAE papers) that the PoE approach is unstable during training. Do the authors find a way to overcome this? (There is a solution discussed in Section 2.1, but is not used). And what is the motivation for using PoE over MoE? Further, why do most of the derivations utilize the MoE approach when it is discarded for the PoE approach at the end? I think clarifying these questions would make the narrative of the paper more easy to follow.

Correctness: The claims largely appear to be correct, although I do have some concerns. My first concern is with the proof of Lemma 2, that the new objective lower bounds the usual ELBO (using just the KL divergence). It is certainly not obvious to me how the first line of Equation 9 is equivalent to Equation 6. Is it meant to be a lower bound of the original lower bound in Equation 6 (and by extension lower bounding the log-likelihood itself)? Looking at each respective version, it would appear to be a valid lower bound if KL(q_j || p_moe) >= KL (q_j || p). However, it is not clear to me that this is true. Perhaps I am missing an algebraic trick, but even if so (and the proof is correct) I think it is important that more clarity is added, since this is a central result of the paper. My second concern is with the use of the Product-of-Experts as the "dynamic prior". The authors refer to Nielsen as evidence that the Jensen-Shannon divergence can be written utilizing a wider range of mean functions than just the arithmetic mean. This is fine, but it seems like the proof of Lemma 2 is dependent on the use of the arithmetic mean function (i.e. the Mixture-of-Experts) to aggregate the posteriors? Therefore it appears like this is not necessarily a valid lower bound when using the PoE prior. If there is a valid extension to the proof which admits the use of the geometric mean (as is required for the PoE prior), then it is crucial that is provided, since the PoE prior is the one actually used in all the experiments. Edit: after author rebuttal I realise that Lemma 2 is fine (and relatively trivial).

Clarity: As per my previous points, I do have concerns about the clarity of some of the key proofs in the paper. The definition of the Jensen-Shannon divergence between >2 distributions does not have a reference, and is not readily available upon searching. Indeed, the usual definition seems to use a slightly simpler version, in which the f_M function is always a convex sum. If the f_M function always simply represents a convex sum of its arguments then why not simply explicitly write this? Else if it does represent a more general combination (such as allowing a product) then I think a reference is required. The proof of Lemma 1 in the supplementary materials relies on the reader making the connection that the KL divergence is an f-divergence (and in fact that the authors are talking about the f-divergence at all). This should be made clearer.

Relation to Prior Work: The discussion to prior work appears to the best of my knowledge to be thorough.

Reproducibility: Yes

Additional Feedback: I think that if my questions regarding the validity and clarity of some of the central theoretical claims of the paper are answered, then I would be happy to upgrade my score. Edit: I have upgraded my score from 5 to 6 after the author's response, which clarified a couple of my points of confusion.


Review 3

Summary and Contributions: This paper solves the problem of learning a VAE using a mixture of Gaussian as the posterior approximation. This choice of posterior is motivated by applications with multiple modalities, where each mixture component is the posterior under available input modalities. Previous work rely on importance sampling, or IPM/GANs to compute the ELBO, while this paper proposes a new ELBO that can be directly computed.

Strengths: The new ELBO is indeed an interesting contribution. Mixture distributions are certainly very useful in many applications, even when there is a single data modality. The authors show that the ELBO is still correct (i.e. it is still a lower bound to the NLL) and has an extra interpretation as regularization with the Jensen Shannon divergence.

Weaknesses: As I commented in the writing section, the writing is somewhat confusing, and can lead to mis-understanding of the key contribution of the paper. I think the experiments can be strengthened. I think there are two concerns: the log likelihood numbers for continuous distributions are not meaningful for certain datasets. For example, because the MNIST is always black on the borders, a model can achieve infinite likelihood just by predicting with probability one that the border pixels are black. Another way to avoid using the same unimodal posterior for all data modalities is to separate the latent variable into two parts, one shared and one private. I think there are a few experiments in the literature (e.g. JMVAE). Does that solve the problem proposed in the paper? I think some discussion / comparison definitely helps.

Correctness: I did not spot any issues. Error bars would be certainly be good to have in the experiments since the differences are often not very big.

Clarity: I think the structure of the paper can be slightly improved. It was confusing when I read section 3.2. Are you defining a new generative model (with this new prior), or a new ELBO to the original generative model (in Eq. (3))? I think it would also help to define MoE approximation explicitly since that is a key problem the paper tries to solve, currently it’s verbally described in the related work section and easy to miss. The paper would be much easier to read if you explicitly define the MoE, previous instantiation of the ELBO, what the problems are, and how your new ELBO fixes those problems. I think I understood what the authors are trying to say eventually, but it took some effort and rereads.

Relation to Prior Work: I did not find any major missing literature

Reproducibility: Yes

Additional Feedback:


Review 4

Summary and Contributions: The paper proposes MMJSD, a new method for training multimodal generative models based on the generalized Jensen-Shannon divergence. A new lower-bound to the ELBO is derived using a "dynamic prior". The dynamic prior is either a mixture or a product distribution of the different variational posteriors and a static prior (e.g., standard Gaussian). The authors claim that the dynamic prior encourages learning of a shared latent space. Further, the latent space of the individual modalities is factorized into a modality-specific and a shared component. Two multimodal experiments have been designed: MNIST-SVHN-Text (augmenting the MNIST-SVHN multimodal experiment with an additional text modality) and Bimodal CelebA (augmenting CelebA with additional text modality based on the image attributes). Comparisons with competing methods (MVAE and MMVAE) demonstrate an improved performance in terms of cross-modal conditional generation and information gain from the inclusion of the different modalities.

Strengths: The paper represents an interesting contribution in the line of multimodal latent variable generative models. Competing methods such as MVAE [1] and MMVAE [2] seek to "combine" the variational posteriors to encourage information sharing. Instead, MMJSD aggregates the information via a "dynamic prior" which I find interesting. The key idea is simple and the authors have shown that the proposed objective is a lower bound of the ELBO and consequently of the marginal log-likelihood. The experiments demonstrate that the MMJSD works well in practice can leverage information from multiple sources better than MVAE and MMVAE. [1] Wu, Mike, and Noah Goodman. "Multimodal generative models for scalable weakly-supervised learning." Advances in Neural Information Processing Systems. 2018. [2] Shi, Yuge, et al. "Variational mixture-of-experts autoencoders for multi-modal deep generative models." Advances in Neural Information Processing Systems. 2019.

Weaknesses: An important feature of a generative model (multimodal or not) is its ability to generate new random samples. Unfortunately, the quality of random generations from MMJSD is poor. Table 3 shows that the MMJSD is worse than MMVAE and significantly worse than MVAE in terms of the quality of randomly generated samples. In general, the sample quality of MMJSD is considerably worse than MVAE, unless modality-specific latent spaces are used. I suspect that the poor quality of random samples can be attributed to the fact that the authors use a dynamic prior for training and only the static component p(z) is used for random generation. I suggest that the authors discuss this problem and possible solutions in the main text. Although the authors design a trimodal experiment, the third modality is a bit simplistic. A stronger experiment, perhaps with more number of modalities, is needed to demonstrate the scalability and accuracy of MMJSD. In general, the experiments section could be improved by adding more experiments (e.g., CUB-Captions in MMVAE, Computer Vision case study in MVAE).

Correctness: The method and theory are technically sound. Line 48-49: The authors claim that they are the first to perform experiments with more than 2 modalities. This is not entirely correct. See section 6 in [1] where MVAE is trained on 6 modalities in total. [1] Wu, Mike, and Noah Goodman. "Multimodal generative models for scalable weakly-supervised learning." Advances in Neural Information Processing Systems. 2018.

Clarity: The manuscript is well-written. Following are some questions/suggestions that may help improve the clarity of the text: * Since the authors use the PoE form of the dynamic prior in practice (line 168), it should be discussed in more detail. Lemma 2 discusses the MoE prior, but a similar result for PoE is missing. * How the baselines were augmented with modality-specific latent spaces should be discussed in more detail. * Can the authors clarify how the Generation Coherence was evaluated for random generation? (Table 2)

Relation to Prior Work: The related works and their corresponding limitations are discussed sufficiently. The similarities and differences of the MMJSD objective to the MVAE when using the PoE dynamic prior must be discussed briefly.

Reproducibility: Yes

Additional Feedback: Some suggestions to improve the paper: * Additional experiments to demonstrate the scalability and accuracy of the proposed method. * A discussion of the quality of random generation and potential solutions to the corresponding limitation in MMJSD. Questions: * Do the authors have a hypothesis for the dip in generation coherence around 0.4 weight for the static prior (Fig. 4b in appendix)? * One would expect the random generation coherence to decrease with decreasing weight for the static prior since random samples are generated via the static prior. However, such behavior is not seen in Fig. 4b. Do the authors have any thoughts on this? [Post Rebuttal] I thank the authors for clarifying on the above questions. I am keeping my score.

[Author Response · NeurIPS 2020]

1  We would like to thank all reviewers for their detailed, thoughtful and valuable feedback. We are encouraged that
2  the reviewers are convinced by our motivation [R1, R2], methodical contribution [R3, R4], experimental results [R1,
3  R2] and the thorough comparison to related work [R2, R3, R4]. We address the reviewers' comments below by
4  first clarifying the derivation of our proposed ELBO, followed by a detailed explanation of certain aspects of our
5  experimental setup. Lastly, we address individual questions.

**Dynamic Prior and generalized JS-Divergence**. [R1, R2, R4] There are two major reasons for using a PoE as
distribution for the dynamic prior: 1) The KL-divergence between a Gaussian distribution and a PoE of Gaussians
can be calculated in closed-form, as mentioned in Section 3.4. 2) PoE-based models [27] are able to approximate the
joint posterior distribution well which we would like to utilize. Instabilities in training and overconfident experts as
mentioned in [21] are mainly due to difficulties in the optimization of unimodal posterior approximations. In our case,
the JS-divergence allows us to optimize the unimodal and multimodal approximation functions jointly, leading to a
stable training of the PoE approach. The fact that the standard JS-divergence is defined via mixture distribution was the
main reason to use the MoE in the derivations. As such, the derivation based on the MoE is a special case, which is
most familiar to readers, but the result is more general and holds for any abstract mean distribution (as in [18]). We will
state this more clearly in the final version of the paper.
[R2, R3] Derivation of ELBO (Eq. 9): Similar to [24], the dynamic prior is defined by a data-dependent function. In the
general formulation of Eq. (6), there are not yet any additional assumptions on the prior distribution. The dynamic
prior defines a valid distribution for the MoE- as well as for the PoE-variant (see appendix B.2 and B.3) and hence is
a well-defined prior. This makes the first line of Eq. (9) a valid ELBO - independent of the exact formulation of the
dynamic prior as long as it is a proper distribution. We would like to emphasize that Eq. (9) is not meant to be a lower
bound to Eq. (6). As stated in Section 3.2 and 3.3 and proven with Lemma 2, we only claim the validity of the proposed
ELBO using the JS-divergence. As proven in [18], the derivation can be generalized to any abstract mean distribution,
including the geometric mean that defines the PoE. We will point this out in the final version of the paper. [R2] The
references for the JS-divergence for $M$ distributions [1,14] are given in the introduction, we will add them in Section 3
as well. The reference for the extension to generalized means in [18] is given in Section 3.4.

**Experiments**. [R1, R2] We highlight the advantage of modality-specific (MS) subspaces using conditional generation
plots (cf. Figure 1) instead of latent traversals. For conditional generation, MS subspaces allow to mix and match
different shared and MS encodings. The columns show that the shared and MS spaces disentangle (every row is a
different random sample from the MS latent subspace). The shared information (digit number) is invariant per column,
while the MS information is invariant per row. This gives empirical evidence that MS and shared latent spaces encode
different information. We will include a visualization of low-dimensional embeddings of the shared and MS latent
spaces in the Appendix. [R4] We already describe the details of all the models incl. MS spaces in the Appendix (Section
C.2.2). To this, we will add the respective ELBOs utilizing MS subspaces. [R3, R4] To the best of our knowledge,
we are the first to perform an experiment with three different types of modalities. In our opinion, different modalities
should contain information that is specific to each modality. In [27]'s vision study, the different modalities are filtered
versions of the original modality which prevents them from having true modality-specific information.
[R4] In our opinion, the quality of generated samples is only one side of the coin to evaluate multi-modal generative
models. We are convinced that only its combination with the coherence of generated samples allows for a valid
assessment. Although the MVAE model is able to generate high quality samples, comparing Table 3 and 4 shows
that the quality of samples comes at the cost of reduced coherence accuracy (for conditional generation) which is
significantly lower than MMVAE's and ours. Additionally, by introducing MS subspaces, we find a solution to generate
samples of high quality which are coherent between modalities - random generation and all subsets of samples. [R3] We
report the NLL-numbers for completeness as it is a de-facto standard evaluation method for VAE-based models despite
the known weaknesses. [R1] The introduction of modality-specific subspaces leads to a small overhead regarding the
hyperparameters (incl. priors). In our experiments, we used standard Gaussian priors - as it is common for VAE-based
models - for the modality-specific subspaces which work very well in practice. [R3] A comparison of runtimes can be
found in the Appendix (Table 8, Section C.2.4) which highlights the inefficiency of IWAE-based models (3x longer
training time) as mentioned in the related work Section.

**Further Questions.** [R3] "Comparison to JMVAE": The introduction of the JS-divergence has a similar motivation as
JMVAE [22] which we discuss in the related work. By using the JS-divergence the unimodal posterior approximations
are automatically optimized for being close to a joint posterior distribution (the dynamic prior in our case). We are able
to solve this in a scalable way while [22] have to use an exhaustive approach. [R4] "Ablation studies in the Appendix":
Our hypothesis for the stable random coherence in this experiment is that the unimodal posteriors are learned in a way
that their mean distribution is similar to static prior (we have no final hypothesis for the dip yet). [R4] "Calculation
of coherence for random generation": The generated samples were evaluated using a pre-trained classifier for each
modality. If all modalities show the same content/shared information, this is a coherent generation. From there, we
calculate accuracy/precision recall.

[Meta-Review · NeurIPS 2020]

This paper proposes a new method for training multimodal generative models based on Jensen-Shannon Divergence. At the beginning, it received 5666 scores; after rebuttal, the scores have been increased to 6666, which are 4 weak accept recommendations. All the reviewers agree that this paper is well organized and the main techniques are also clearly presented. Therefore, the AC recommends accepting the paper. The reviewers also gave a lot of more detailed comments. The authors are encouraged to use these comments to further improve the paper.